# ParaTool: Shifting Tool Representations from Context to Parameters

**Zekai Yu**[1]   **Qi Meng**[1]   **Qizhi Chu**[1]   **Yu Hao**[1]   **Chuan Shi**[1]   **Cheng Yang**[1]

## Abstract

Tool calling extends large language models (LLMs) by enabling grounded interaction with external executable interfaces, thereby supporting environment-coupled problem solving. However, mainstream in-context learning (ICL) approaches typically incorporate detailed tool documentation and usage examples directly into the context. This results in substantial inference overhead and heightened risks of hallucination as the context length grows. Conversely, while tuning-based methods improve general tool-calling capabilities, they often fail to effectively internalize the specific details of previously seen tools, thereby retaining a dependency on in-context documentation. To address these limitations, we propose *ParaTool*, a framework that projects each tool into a dedicated, loadable set of parameters. By equipping a dynamic integration of these parameterized tools, the LLM can perform tool calling without relying on in-context documents or examples. Specifically, our approach consists of three stages: (1) *parametric tool pre-training* encapsulates the knowledge of different tools into independent parameter modules; (2) *soft tool selection* employs a gating network to dynamically weigh and aggregate relevant tool parameters; and (3) *parametric tool fine-tuning* jointly updates tool parameters to align the training and inference processes. Experiments on Stable ToolBench and BFCL demonstrate that *ParaTool* significantly outperforms strong ICL-based baselines, achieving superior performance while reducing computational complexity.

**Code:** https://github.com/BUPT-GAMMA/ParaTool

## 1. Introduction

Despite demonstrating remarkable capabilities in natural language understanding and complex reasoning, large language models (LLMs) (Wei et al., 2022a;b; Bubeck et al., 2023) remain constrained by the isolation from external environments (Schick et al., 2023; Mialon et al., 2023). To

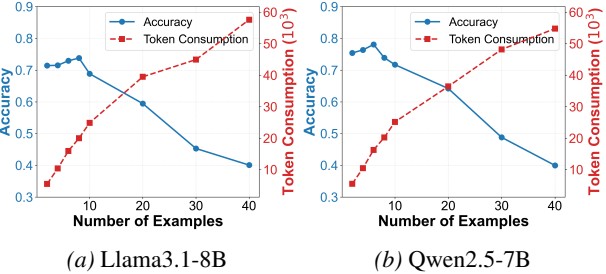

*(a)* Llama3.1-8B          *(b)* Qwen2.5-7B

*Figure 1.* We incorporate tool documents and usage examples into the prompt to facilitate ICL-based tool calling. However, contrary to the expectation that more context yields better results, we observe that tool calling accuracy peaks and then drops as the number of examples increases.

bridge this gap, tool calling has emerged as a transformative paradigm. By empowering models to invoke executable interfaces (Patil et al., 2024; Qin et al., 2024b), this approach evolves LLMs from text generators into autonomous agents (Sumers et al., 2023; Wu et al., 2024), enabling them to solve complex tasks through grounded real-world interaction. In typical scenarios, LLMs are often required to select and execute the appropriate tool from a given toolset, based on specific contexts (Qin et al., 2024a; Patil et al., 2025; Qin et al., 2024b). Mainstream tool-use paradigms predominantly rely on in-context learning (ICL) (Yao et al., 2022; Lu et al., 2023), which typically integrates detailed tool documentation and usage examples directly into the input prompt to provide the model with relevant tool knowledge. However, as illustrated in Figure 1, as we introduce an increasing number of usage examples into the context, the model's tool-calling capability paradoxically declines. The extensive context resulting from documentation and examples not only incurs high inference latency and memory overhead but also exacerbates the risk of hallucination, making it difficult for the model to precisely capture effective content within verbose prompts. On the other hand, by injecting tool-use knowledge in massive tool datasets directly into model parameters, tuning-based methods can effectively enhance the model's generalizability in following tool-use instructions (Qin et al., 2024b; Patil et al., 2024). But due to the vast scale and diversity of tools involved in training, such models often fail to internalize specific details of previously seen tools, and remain reliant on external documentation during inference.

---

[1]Beijing University of Posts and Telecommunications, Beijing, China. Correspondence to: Cheng Yang <yangcheng@bupt.edu.cn>.

*Proceedings of the $43^{rd}$ International Conference on Machine Learning*, Seoul, South Korea. PMLR 306, 2026. Copyright 2026 by the author(s).

To address these challenges, we propose a novel method named *ParaTool*, where each tool is projected into a dedicated set of parameters. By dynamically loading a weighted aggregation of these parameterized tools, the LLM can then perform tool calling without relying on in-context documents or examples. Specifically, our approach consists of three stages: (1) **Parametric Tool Pre-training**: We construct pre-training tasks derived from tool documentation to encode each tool into an independent, loadable parametric representation; (2) **Soft Tool Selection**: A gating network is trained to dynamically weigh candidate tools according to the specific context; (3) **Parametric Tool Fine-tuning**: We provide the LLM with the weighted aggregation of parametric tools and jointly optimize the tools' parameters, aligning the training and inference reasoning processes. This framework thus enables the LLM to master different tools in a plug-and-play manner. In summary, this paper makes the following key contributions:

• We propose a novel paradigm that enables LLM tool calling without relying on in-context documents or examples, based instead on dynamic integration of parameterized tools.

• We develop a three-stage framework comprising parametric tool pre-training, soft tool selection, and parametric fine-tuning, which collectively enable the LLM to master tools in a plug-and-play manner.

• Extensive experiments demonstrate that *ParaTool* significantly outperforms ICL-based baselines, achieving relative improvements of $9.71\%$ in pass rate on Stable ToolBench and $4.22\%$ in accuracy on BFCL. The approach also reduces computational complexity (measured in FLOPS) by up to $92.22\%$ on Stable Toolbench and $94.45\%$ on BFCL.

## 2. Related Work

### 2.1. Tool Learning

Tool learning aims to empower LLMs to interact with external interfaces, extending their capabilities beyond static internal knowledge to accomplish complex real-world tasks (Yao et al., 2022; Schick et al., 2023). Current approaches generally fall into two paradigms: In-Context Learning (ICL) methods integrate detailed tool documentation and usage examples directly into the input prompt to guide the model in mastering tool usage (Wei et al., 2022b; Hsieh et al., 2023; Paranjape et al., 2023; Du et al., 2024; Shi et al., 2024; Xu et al., 2024; Yuan et al., 2025). For example, Easy-Tool optimizes this by condensing original documentation into high-quality guidelines and incorporating several usage examples (Yuan et al., 2025). In contrast, tuning-based methods inject general tool-calling capabilities directly into parameters (Hao et al., 2023; Yang et al., 2023; Wang et al., 2024; Zhang et al., 2025; Prabhakar et al., 2025). For example, ToolLLaMA fine-tunes LLaMA on massive instruction-

solution pairs to master diverse APIs (Qin et al., 2024b) and ToolAlign aligns general tool-use behaviors with helpfulness, harmlessness, and autonomy through instruction tuning and preference optimization (Chen et al., 2024b). However, ICL suffers from substantial inference overhead and heightened hallucination risks caused by processing verbose documentation (Xu et al., 2024). Conversely, while current tuning-based methods can improve general instruction adherence, they fail to internalize specific details of previously seen tools and remain dependent on in-context document details to generate tool callings (Qin et al., 2024a).

### 2.2. Parameter-Efficient Fine-Tuning

Parameter-Efficient Fine-Tuning (PEFT) has emerged as a widely adopted paradigm for adapting LLMs to downstream tasks while mitigating the prohibitive computational costs of full-parameter training (Houlsby et al., 2019; Li & Liang, 2021; Liu et al., 2022; Lialin et al., 2023). Techniques like Low-Rank Adaptation (LoRA) (Hu et al., 2022) have become the standard paradigm for efficient capability injection in various domains (Zhang et al., 2023; Liu & Low, 2023; Zhou et al., 2024; Su et al., 2025). For instance, Platypus leverages LoRA to efficiently enhance logical reasoning capabilities (Lee et al., 2023). PEFT is also widely explored in tuning-based tool learning methods (Qin et al., 2024a). Existing work usually adopts PEFT by training a single adapter over all tools to improve general tool-calling capabilities. In contrast, *ParaTool* represents a paradigm shift towards tool-level parameterization.

## 3. Methodology

### 3.1. Problem Formalization

In this subsection, we will first introduce the tool-use task of LLMs. Then we will formalize the context-based and parameter-based paradigms of tool representations.

**Tool-use Task.** Typically, the task requires an LLM to iteratively select and execute a proper tool from a given toolset based on query and historical context. Formally, given a query $q$ and a toolset $\mathcal{T} = \{T_i\}_{i=1}^{|\mathcal{T}|}$, the goal of an LLM at the $k$-th step is to generate action $a_k$ by

$$f_\theta(a_k | q, \mathcal{H}_{k-1}, \mathcal{T}), \qquad (1)$$

where action $a_k$ includes a selected tool as well as its arguments, $f_\theta$ is the LLM with parameters $\theta$, and $\mathcal{H}_{k-1} = (a_1, o_1, a_2, o_2 \dots a_{k-1}, o_{k-1})$ concatenates previous actions and observations (*e.g.,* environmental feedback).

**Context-based Tool Representation.** To provide the LLM $f_\theta$ with necessary tool information, previous methods usually concatenate the documents $T_i^{Doc}$ and examples $T_i^{Exp}$ of each tool $T_i$ into the prompt, leading to a relatively long

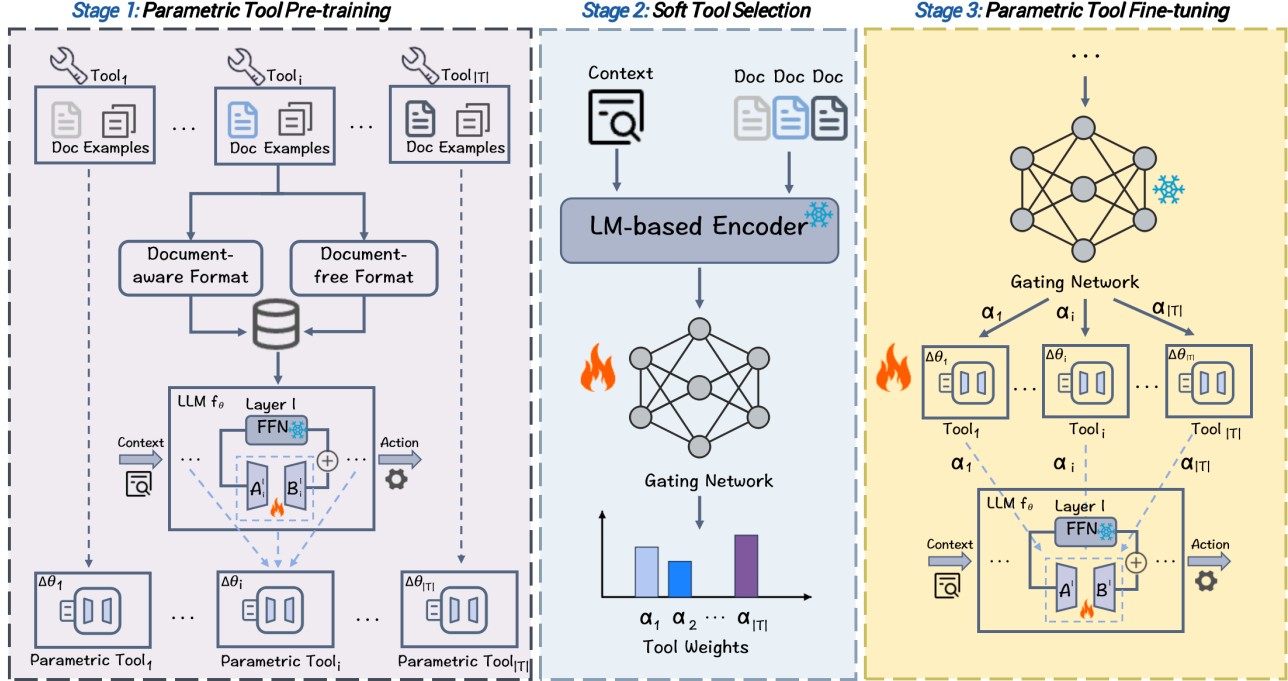

*Figure 2.* The three-stage pipeline of our proposed *ParaTool*: (1) Parametric Tool Pre-training, where each tool is independently transformed into corresponding parametric representation; (2) Soft Tool Selection, which trains a gating network to predict aggregation weights for the soft composition of tool representations; and (3) Parametric Tool Fine-tuning, where the tool-specific parameters are jointly fine-tuned with the predicted weights to align the training and inference processes.

context. Then the action generation can be written as

$$f_\theta(a_k|q, \mathcal{H}_{k-1}, \bigoplus_{i=1}^{|\mathcal{T}|} T_i^{Doc}, \bigoplus_{i=1}^{|\mathcal{T}|} T_i^{Exp}), \quad (2)$$

where $\bigoplus$ denotes the concatenation operation.

**Parameter-based Tool Representation.** In this paper, we propose to transform each tool $T_i$ into a small amount of model parameters $\Delta\theta_i$ for the LLM instead. At the $k$-th step of action generation, we will dynamically assign aggregation weight $\alpha_i^k \in \mathbb{R}$ for each tool $T_i$, and predict without explicit documents or examples in the prompt:

$$f_{\theta+\Delta\theta^k}(a_k|q, \mathcal{H}_{k-1}), \quad (3)$$

where $\Delta\theta^k = \sum_{i=1}^{|\mathcal{T}|} \alpha_i^k \Delta\theta_i$. The parameter-based tool representation allows the LLM to utilize tools efficiently with much shorter context.

### 3.2. Framework Overview

As shown in Figure 2, we design a three-stage pipeline for the proposed parameter-based paradigm.

**Parametric Tool Pre-training**: Transforming each tool into corresponding LLM parameters based on its documents and examples independently.

**Soft Tool Selection**: Training a gating module by next tool prediction to obtain the aggregation weights for soft composition of tool representations.

**Parametric Tool Fine-tuning**: Jointly fine-tuning the parameters of different tools with predicted aggregation weights to align the inference process.

### 3.3. Parametric Tool Pre-training

In this subsection, our objective is to encode the knowledge of every tool $T_i \in \mathcal{T}$ into a parametric representation $\Delta\theta_i$. This representation is designed to empower the LLM with the capability to effectively invoke $T_i$ without explicit documents or examples in the context.

Firstly, we construct a dedicated dataset $\mathcal{D}_i$ for each tool $T_i$, comprising a collection of tool-calling traces with $T_i$ as the next tool to be predicted. In these datasets, each trace is processed into two distinct training instances:

*Document-aware Formatting*: The input prompt includes the full tool documentation and descriptions. This provides rich semantic context to help initialize the tool's parametric representation.

*Document-free Formatting*: We strip away the documentation and descriptions. This forces the LLM to rely on the parametric representation rather than the context.

Subsequently, we leverage the collected dataset $\mathcal{D}_i$ to optimize the parametric representation $\Delta\theta_i$. For the above two formats, the LLM predicts as $f_{\theta+\Delta\theta_i}(a_k|q, \mathcal{H}_{k-1}, T_i^{Doc})$ and $f_{\theta+\Delta\theta_i}(a_k|q, \mathcal{H}_{k-1})$, respectively. To enable the LLM to generate correct tool calls, we optimize the log-likelihood of generating all tokens in each target action $a_k$.

For detailed implementation, we adhere to the Low-Rank Adaptation (LoRA) paradigm (Hu et al., 2022), and instantiate $\Delta\theta_i$ as a set of low-rank matrices targeting the Feed-Forward Network (FFN) weights of the backbone LLM. This architectural design isolates specific trainable parameters for each tool $T_i$, enabling the model to master different tools in a plug-and-play manner. Formally, assume that the LLM backbone has $L$ layers of FFNs, the parametric representation $\Delta\theta_i = \{A_{i,l}B_{i,l}^\top\}_{l=1}^L$, where $A_{i,l}$ and $B_{i,l}$ are low-rank matrices for the $l$-th layer. We accordingly train the low-rank matrices $\{A_{i,l}, B_{i,l}\}_{l=1}^L$ related to $\Delta\theta_i$ with the instances in $\mathcal{D}_i$. This ensures that the functional semantics are encoded solely into the lightweight parameters.

### 3.4. Soft Tool Selection

At the inference phase, a hard selection strategy can pick the most suitable tool and equip its corresponding parameters for action generation. But the tool use performance will heavily rely on the accuracy of tool selection, thereby harming the robustness of the entire framework. To mitigate this issue, we introduce a soft tool selection mechanism that assigns aggregation weights to different tools based on the current context.

Formally, we first encode the context $(q, \mathcal{H}_{k-1})$ into $c_k$ with a frozen language model-based encoder $\mathrm{Enc}(\cdot)$ (e.g., the last token embedding). Similarly, we encode the document of each tool $T_i$ and obtain its vector $d_i$. Then these embeddings are projected via a multi-layer perceptron (MLP) to compute relevance scores, which are then normalized via softmax to yield the weights $\alpha^k \in \mathbb{R}^{|\mathcal{T}|}$:

$$\alpha_i^k = \mathrm{softmax}(\mathrm{MLP}(c_k, d_i, c_k \odot d_i, |c_k - d_i|)), \quad (4)$$

where $\odot$ denotes the element-wise product.

The training objective of the above gating network is the standard cross-entropy loss between tool weights $\alpha^k$ and the tool used in the ground truth action $a_k$. To prevent the gate from collapsing into a hard selector, we introduce an entropy regularization term $-\lambda \cdot H(\alpha^k)$ with hyper-parameter $\lambda$, encouraging softer weights for tool composition.

### 3.5. Parametric Tool Fine-tuning

Note that the pre-trained tool parameters are learned only based on its own trace dataset. To align with the inference process, we freeze the gating network and jointly fine-tune the tool parameters under the soft tool composition.

Formally, we first aggregate the tool parameters $\Delta\theta^k = \sum_{i=1}^{|\mathcal{T}|} \alpha_i^k \Delta\theta_i$, and then predict next action as Eq. (3). In practice, we only preserve the $N$ candidate tools provided by the user or the tools with largest weights, where $N \ll |\mathcal{T}|$. Thus, the calculation of parameter fusion does not need to sum over $|\mathcal{T}|$ terms. Similar to the pre-training stage, we then optimize the matrices by maximizing the log-likelihood of generating all tokens in each target action $a_k$.

The fine-tuning stage can align the training dynamics with the inference phase, where multiple parametric tools are simultaneously activated and superimposed onto the backbone parameters. This forces the parametric representations to adapt to the induced weight distribution, learning to coordinate their updates to maximize collective utility.

### 3.6. Theoretical Analysis of Soft Tool Composition

In this subsection, we prove that our soft tool composition strategy is more robust than a hard selection one based on the concept of certified robustness radius (Zhai et al., 2020).

To simplify the notations in theoretical analysis, we let $x$ denote the input context (comprising query $q$ and history $\mathcal{H}$), and $y$ denote the target action. We define the soft-composed model's loss function as $\mathcal{J}_\alpha(x, y)$, computed by the LLM $f$ with parameters $\theta + \sum_{i=1}^{|\mathcal{T}|} \alpha_i \Delta\theta_i$, where $\alpha \in \Delta_{|\mathcal{T}|-1}$ denotes the aggregation weights with their sum equals to 1. Let $g_\alpha(x, y) \triangleq \nabla_x \mathcal{J}_\alpha(x, y)$ denote the input gradient of the soft-composed model. Similarly, let $g_i(x, y)$ denote the gradient derived when the model exclusively uses tool $T_i$ (i.e., parameters $\theta + \Delta\theta_i$).

**Definition 3.1** (Robustness Radius). The local robustness at input context $x$ is measured by the loss-based radius $R_\varepsilon(x, y; \alpha)$, defined as:

$$R_\varepsilon(x, y; \alpha) \triangleq \sup \Big\{ z \geq 0 : \forall \delta, \|\delta\|_2 \leq z, $$
$$\mathcal{J}_\alpha(x + \delta, y) \leq \mathcal{J}_\alpha(x, y) + \varepsilon \Big\}. \quad (5)$$

**Assumption 3.2** (Local Smoothness). The loss function $\mathcal{J}_\alpha$ is locally $\beta$-smooth. Specifically, for a small perturbation $\delta$:

$$\mathcal{J}_\alpha(x + \delta, y) \leq \mathcal{J}_\alpha(x, y) + \langle g_\alpha(x, y), \delta \rangle + \frac{\beta}{2}\|\delta\|_2^2. \quad (6)$$

In this paper, we investigate the robustness radius based on local gradients. In Appendix A, we show that under Assumption 3.2, a smaller gradient norm implies a larger lower bound on the robustness radius.

**Definition 3.3** (Gradient Alignment Coefficient). To quantify the gradient conflict across the parameter representations of different tools, we define the alignment coefficient $\rho_g \in [0, 1]$ as the upper bound of cosine similarity:

$$\rho_g \triangleq \max \Big\{ 0, \sup_{i \neq j} \sup_{(x,y)} \cos(g_i(x, y), g_j(x, y)) \Big\}. \quad (7)$$

Intuitively, when the gradients of different tools are not perfectly aligned (*i.e.,* $\rho_g < 1$), the soft tool composition strategy can enhance stability by exploiting the cancellation effect among gradients. Now we formally discuss how $\alpha$ affects the gradient norm.

**Assumption 3.4** (Bounded Gradients). For all tools $T_i \in \mathcal{T}$, the gradient norm is bounded: $\|g_i(x,y)\|_2 \leq G$.

**Assumption 3.5** (Linear Aggregation with Bounded Residual). The gradient of the composed model approximates the weighted sum of individual tool gradients:

$$g_\alpha(x,y) = \sum_{i=1}^{|\mathcal{T}|} \alpha_i g_i(x,y) + \xi_\alpha(x,y), \qquad (8)$$

with the residual bounded by $\|\xi_\alpha\|_2 \leq \delta_g$.

**Theorem 3.6** (Gradient Norm Bound). *Under Assumptions 3.4 and 3.5, for any aggregation weights $\alpha$, we have:*

$$\|g_\alpha(x,y)\|_2 \leq G\sqrt{\rho_g + (1-\rho_g)\|\alpha\|_2^2} + \delta_g. \quad (9)$$

**Corollary 3.7** (Robustness of Soft Composition). *Suppose Assumptions 3.2, 3.4 and 3.5 hold. A softer composition (characterized by a lower $\ell_2$-norm of weights $\|\alpha\|_2^2$) strictly decreases the upper bound on the gradient norm $\|g_\alpha(x,y)\|_2$. Consequently, soft composition yields a strictly larger lower bound of robustness radius $R_\varepsilon(x,y;\alpha)$ compared to the hard selection (where $\|\alpha\|_2^2 = 1$).*

This confirms that soft tool composition theoretically improves the lower bound of local robustness (see Appendix A for detailed proof).

### 3.7. Complexity Analysis

**Context-based Complexity.** For a standard Transformer layer, the computational complexity is typically expressed as $\mathcal{O}(S^2 h + Sh^2)$, where $S$ is the sequence length and $h$ is the hidden size. We omit the output length as it is much smaller than the input one in the tool-use scenario. Here $\mathcal{O}(S^2 h)$ corresponds to the quadratic self-attention mechanism, and $\mathcal{O}(Sh^2)$ to the linear Feed-Forward Networks (FFN). Following the formulation above, the context-based paradigm concatenates all tool documents and examples into the input, resulting in a total length $S_{ctx} = |q| + |\mathcal{H}_{k-1}| + \sum_{i=1}^{|\mathcal{T}|}(|T_i^{Doc}| + |T_i^{Exp}|)$. Evidently, as $|T_i^{Doc}|$ and $|T_i^{Exp}|$ increase rapidly, this massive static context leads to a drastic explosion in computational overhead.

**Parameter-based Complexity.** In our parameter-based framework, while the input sequence is reduced to $S_{par} = |q| + |\mathcal{H}_{k-1}|$, additional computational overhead arises from the dynamic aggregation of $N$ LoRA experts. Note that we do not have to calculate the product of low-rank matrices $AB^\top$ by altering the order of matrix multiplications. Thus,

the overhead can be reduced to $\mathcal{O}(S_{par}NLhr)$, where $L$ is the number of FFN layers and $r$ is the rank of LoRA matrices. In practice, $NLr \approx h$. Thus, the additional overhead does not increase the overall complexity. Compared to the context-based approach, the parameter-based one enjoys less computation by transforming the heavy processing of verbose documents into lightweight low-rank computations.

## 4. Experiments

### 4.1. Datasets

To thoroughly evaluate the effectiveness of *ParaTool*, we conduct experiments on two representative datasets: Stable ToolBench (Qin et al., 2024b) and the Berkeley Function-Calling Leaderboard (BFCL-V2) (Patil et al., 2025). Stable ToolBench encompasses $2,098$ distinct tools and $765$ tasks across six subsets, ranging from single-tool to complex multi-tool instructions. BFCL-V2 features $2,034$ distinct tools and $1,693$ test cases, spanning categories that include `Parallel`, `Multiple`, and the challenging `Parallel Multiple` tasks. The tasks are further divided into Non-live and Live versions, yielding six categories. These tasks have diverse user queries characterized by varying tones, multi-turn interactions, and multilingual inputs. More details about the datasets can be found at Appendix B.

Besides, we collect trace datasets that can be utilized by both context-based and parameter-based methods (see Appendix C for more details):

• For **Stable ToolBench:** We construct examples based on the training set of ToolBench (Qin et al., 2024b). For each target tool, we collect user queries requiring the tool and enforce a decontamination process to prevent data leakage, ensuring that no queries from the Stable ToolBench test set are included. Subsequently, we utilize Best-of-N sampling to generate high-quality trajectory for each query based on GPT-4o. This yields an average of 75 examples per tool.

• For **BFCL:** We begin by extracting and deduplicating tool schemas from the official dataset. Utilizing GPT-4o, we then generate 20 atomic examples for each tool, consisting of user queries and compliant function calls. These atomic examples are composed into complex trajectories varying from single-tool invocations to multi-tool selection tasks.

### 4.2. Experimental Settings

We conduct experiments based on Llama-3.1-8B-Instruct (Grattafiori et al., 2024) and Qwen2.5-7B-Instruct (Team et al., 2024) using 8 NVIDIA A800-40G GPUs. For both pre-training and fine-tuning stages, we configure LoRA with a rank $r = 16$ and a scaling factor $\alpha = 64$. For the gating network, the hidden dimension is set to $512$ and the number of layers is 3. Detailed

*Table 1.* Experimental results across the six categories of Stable Toolbench dataset. The Win Rates are calculated by comparing with the Context+Docs baseline. **Bold** and underlined values indicate the best and second-best performance, respectively.

| | I1-Inst. | | I1-Cat. | | I1-Tool. | | I2-Inst. | | I2-Cat. | | I3-Inst. | | Average | |
|---|---|---|---|---|---|---|---|---|---|---|---|---|---|---|
| | **Pass** | **Win** | **Pass** | **Win** | **Pass** | **Win** | **Pass** | **Win** | **Pass** | **Win** | **Pass** | **Win** | **Pass** | **Win** |
| *Model: Llama3.1-8B-Instruct* | | | | | | | | | | | | | | |
| **Context+Docs** | 51.63 | - | 59.51 | - | 53.80 | - | 58.06 | - | 61.32 | - | 52.46 | - | 56.13 | - |
| **Context+Docs & Examples** | 62.75 | 65.36 | 66.87 | 68.01 | 62.66 | 60.76 | 68.55 | 63.71 | 69.81 | 67.92 | 63.93 | 55.74 | 65.67 | 63.58 |
| **Global Parameterization** | 34.64 | 37.91 | 32.52 | 34.97 | 36.08 | 30.38 | 41.94 | 38.71 | 36.79 | 32.08 | 29.25 | 21.31 | 35.20 | 32.56 |
| **ToolLLaMA** | 64.42 | 62.58 | 68.63 | 71.24 | 61.11 | 64.55 | 70.75 | 67.92 | 66.94 | 69.35 | 60.66 | 57.38 | 65.41 | 65.50 |
| **Parameter+Embsim** | 54.25 | 56.86 | 60.74 | 63.90 | 55.06 | 55.70 | 62.10 | 64.52 | 57.55 | 60.38 | 47.54 | 44.26 | 56.21 | 57.59 |
| **Parameter+ToolRetriever** | 60.13 | 56.68 | 63.19 | 66.26 | 59.49 | 61.39 | 65.32 | 64.52 | 61.32 | 62.26 | 50.82 | 52.46 | 60.05 | 60.60 |
| **Parameter+Reinvoke** | 62.09 | 63.40 | 65.64 | 68.01 | 63.92 | 66.46 | 68.55 | 70.16 | 65.09 | 66.98 | 57.38 | 55.74 | 63.78 | 65.13 |
| *ParaTool* (ours) | **67.97** | **69.93** | **69.94** | **73.01** | **67.09** | **69.62** | **74.19** | **75.00** | **72.64** | **74.53** | **65.57** | **68.85** | **69.57** | **71.82** |
| *Model: Qwen2.5-7B-Instruct* | | | | | | | | | | | | | | |
| **Context+Docs** | 70.49 | - | 58.28 | - | 55.09 | - | 51.00 | - | 54.72 | - | 58.82 | - | 58.06 | - |
| **Context+Docs & Examples** | 73.77 | 77.12 | 64.89 | 68.71 | 68.35 | 69.62 | 65.00 | 61.29 | 64.15 | 68.87 | 65.57 | 68.85 | 66.96 | 69.08 |
| **Global Parameterization** | 42.48 | 39.87 | 37.42 | 34.36 | 37.97 | 34.18 | 44.35 | 47.58 | 42.45 | 45.28 | 31.33 | 27.87 | 39.33 | 38.19 |
| **ToolLLaMA** | 72.40 | 68.10 | 70.59 | 75.16 | 65.82 | 71.52 | 66.98 | 63.20 | 68.54 | 72.58 | 65.57 | 70.49 | 68.31 | 70.18 |
| **Parameter+Embsim** | 65.36 | 71.89 | 57.67 | 62.58 | 55.06 | 59.45 | 54.84 | 62.10 | 58.49 | 61.32 | 50.82 | 54.10 | 57.04 | 61.91 |
| **Parameter+ToolRetriever** | 69.28 | 73.62 | 59.90 | 63.80 | 57.59 | 60.13 | 59.68 | 62.90 | 62.62 | 60.38 | 52.46 | 55.74 | 60.26 | 62.76 |
| **Parameter+Reinvoke** | 70.59 | 75.81 | 68.71 | 74.23 | 67.72 | 70.25 | 69.35 | 73.39 | 72.64 | 71.70 | 60.66 | 62.30 | 68.28 | 71.28 |
| *ParaTool* (ours) | **79.09** | **88.23** | **76.07** | **80.98** | **75.95** | **82.28** | **77.42** | **79.03** | **78.30** | **80.19** | **68.85** | **73.77** | **75.95** | **80.75** |

hyperparameters for each training stage (including learning rates, batch sizes, and regularization terms specific to different datasets) are provided in Appendix D.

Following the evaluation protocols of Stable ToolBench, we employ Pass Rate (%) and Win Rate (%) as metrics: Pass Rate calculates the proportion of successfully completed instructions within a limited budget, while Win Rate utilizes an LLM (*i.e.,* GPT-4o) to evaluate and select the superior trajectory between two provided options. For BFCL, we adopt the official Abstract Syntax Tree (AST) accuracy for evaluation. All reported values are averaged over three runs.

### 4.3. Baselines

To validate our proposed method, we compare against seven baselines ranging from standard prompting to advanced tool retrieval strategies:

• **Context+Docs:** We directly concatenate raw tool documents into the context window, relying solely on the model's knowledge for inference without specific fine-tuning.

• **Context+ Docs & Examples :** Besides the tool documents, we also provide the model with the collected examples for each tool to establish a strong in-context learning baseline. The number of provided examples is carefully tuned between $[0, 40]$.

• **Tuning-based Methods:** To assess the necessity of our tool-specific representations, we compare *ParaTool* against two tuning-based methods that encode tool information into a shared parameter space.

- **Global Parameterization:** This baseline replaces our tool-specific parameterization with a single shared LoRA module, which is trained on the entire dataset and applied uniformly across all tools.

- **ToolLLaMA:** A classic and representative tuning-based approach is adopted as the baseline. Specifically, we preprocess the tool-calling traces into both document-aware and document-free formats, and retrain the LoRA variant of ToolLLaMA(Qin et al., 2024b) on Llama-3.1-8B-Instruct and Qwen2.5-7B-Instruct. During inference, to ensure a fair comparison with *ParaTool*, we use its tool retriever to select relevant tools from the full toolset, rather than providing all tool schemas in the context.

• **Retrieval-based Tool Selectors:** To isolate the efficacy of our gating mechanism for tool selection, we replace it with three established retrieval methods while retaining our backbone of tool parameterization:

- **EmbSim:** A dense retrieval method using OpenAI's `text-embedding-3-large`. It selects tools based on the cosine similarity between the query embedding and tool document embeddings.

- **Re-Invoke:** An unsupervised strategy proposed by Chen et al. (2024a) that employs query rewriting and

Table 2. Experimental results across the six categories of BFCL dataset. **Bold** and underlined values indicate the best and second-best performance, respectively. Note that the Parallel categories does not require any tool selection.

| Method | Non-live | | | | Live | | | |
|---|---|---|---|---|---|---|---|---|
| | Parallel | Multiple | Parallel Multiple | Average | Parallel | Multiple | Parallel Multiple | Average |
| *Model: Llama3.1-8B-Instruct* | | | | | | | | |
| **Context+Docs** | 88.33 | 95.00 | 85.50 | 89.61 | 56.25 | 71.23 | 55.56 | 61.01 |
| **Context+Docs & Examples** | 88.50 | 95.17 | 87.00 | 90.22 | 79.17 | 73.88 | 61.11 | 71.39 |
| **Global Parameterization** | 72.50 | 62.67 | 46.83 | 60.67 | 60.42 | 57.99 | 30.56 | 49.66 |
| **ToolLLaMA** | 72.00 | 87.00 | 58.50 | 72.50 | 62.50 | 58.21 | 33.33 | 51.35 |
| **Parameter+Embsim** | - | 91.33 | 63.50 | 81.39 | - | 77.08 | 48.61 | 70.37 |
| **Parameter+Tool Retriever** | - | 92.50 | 64.33 | 82.06 | - | 76.45 | 43.06 | 68.31 |
| **Parameter+Reinvoke** | - | 91.83 | 66.33 | 82.50 | - | 76.48 | 51.39 | 71.10 |
| *ParaTool* (ours) | **89.33** | **95.33** | **89.50** | **91.39** | **85.42** | **78.00** | **65.28** | **76.23** |
| *Model: Qwen2.5-7B-Instruct* | | | | | | | | |
| **Context+Docs** | 88.17 | 95.33 | 84.17 | 89.22 | 58.33 | 75.28 | 62.50 | 65.37 |
| **Context+Docs & Examples** | 88.83 | 96.17 | 87.50 | 90.83 | 75.00 | 77.90 | 69.44 | 74.12 |
| **Global Parameterization** | 72.83 | 83.83 | 53.33 | 70.00 | 58.33 | 47.04 | 31.94 | 45.77 |
| **ToolLLaMA** | 72.17 | 89.50 | 58.33 | 73.33 | 56.25 | 52.14 | 29.17 | 45.85 |
| **Parameter+Embsim** | - | 91.83 | 69.00 | 83.56 | - | 77.02 | 51.39 | 69.19 |
| **Parameter+ToolRetriever** | - | 92.83 | 69.33 | 84.00 | - | 76.95 | 50.00 | 68.71 |
| **Parameter+Reinvoke** | - | 92.50 | 69.67 | 84.00 | - | 76.92 | 56.94 | 71.01 |
| *ParaTool* (ours) | **89.83** | **96.67** | **89.50** | **92.00** | **79.17** | **79.01** | **73.61** | **77.26** |

document expansion to bridge the lexical gap between queries and tool documents.

- **ToolRetriever:** A BERT-based retriever fine-tuned via contrastive learning (Qin et al., 2024b), designed to capture deep semantic relevance for tool selection.

### 4.4. Main Results

As presented in Tables 1 and 2, *ParaTool* consistently achieves state-of-the-art performance across all evaluated models (Llama3.1-8B, Qwen2.5-7B) and diverse benchmark categories: whether in the rigorous format constraints of BFCL or the multi-turn interaction scenarios of Stable Tool-Bench. Notably, on Stable ToolBench, *ParaTool* delivers a significant relative improvement of 5.9% and 11.2% in Average Pass Rate over the strongest baselines for Llama3.1 and Qwen2.5, respectively. For BFCL, the relative improvements are 6.8% and 4.2%, respectively. This consistent superiority empirically validates the feasibility of our proposed method.

Compared to the context-based methods, the advantage of *ParaTool* is particularly pronounced in the `Live Parallel Multiple` category of BFCL, a challenging category for evaluating high-precision, multi-turn tool execution. Specifically, our method outperforms the *Context+Docs* and *Context+Docs & Examples* baselines by relative margins of 17.6% and 6.4%, respectively. Unlike

context-based approaches that rely on passively retrieving information from the context window, our parameter-based paradigm instills tool capabilities as inherent skills, resulting in more accurate invocations.

The performance of *Global Parameterization* highlights the limitations of monolithic representations. For example, its average performance on BFCL is 33% lower than that of the proposed *ParaTool*. By compressing all tool knowledge into a single global set of parameters, the model struggles to disentangle the functionalities of individual tools, which results in severe catastrophic forgetting and tool hallucinations. The results of *ToolLLaMA* further support this observation: *ToolLLaMA* frequently fails to generate correct tool calls due to catastrophic forgetting.

While retrieval-based tool selectors (*e.g.*, *EmbSim*, *Re-Invoke*) perform adequately in single-turn scenarios (such as `Multiple`), they fall short in complex scenarios. For example, in the `Parallel Multiple` task, *ParaTool* outperforms the best tool selector baseline by 31.7%. Compared to our method, these retrievers struggle to analyze the execution feedback chains required in complex tasks.

### 4.5. Ablation Study

In this subsection, we conduct ablation studies to thoroughly validate the design of *ParaTool*. Specifically, we consider

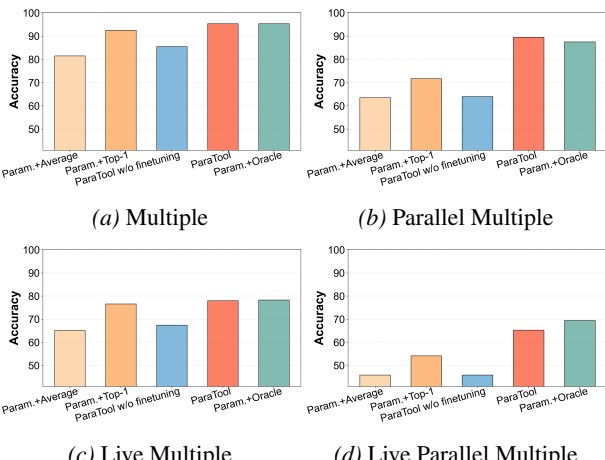

*(a)* Multiple      *(b)* Parallel Multiple

*(c)* Live Multiple      *(d)* Live Parallel Multiple

*Figure 3.* Ablation studies on BFCL dataset with Llama3.1-8B. The Parallel categories are omitted since tool selection is not required.

four variants: *Param.+Average* simply averages the tool parameters; *Param.+Top-1* keeps our trained gating network unchanged while restricting the model to load only the single tool with the highest weight (Top-1) during inference; *Param.+ Oracle* represents the variant where the ground-truth tool parameters are manually activated; and *ParaTool w/o Finetuning* removes the parametric tool finetuning stage. The results on BFCL with Llama3.1-8B are shown in Figure 3 (more results in Appendix E).

Firstly, we verify the necessity of soft tool composition. We can see that the hard selection strategy (*Param.+Top-1*) consistently degrades the accuracy of tool invocation. Especially in the `Live Parallel Multiple` category, the performance drops by nearly 17%. This aligns with our theoretical analysis in Section 3.6. Also, *Param.+Average* performs the worse among different weighting strategies. Thus, a proper weight assignment is necessary for tool parameter aggregation. Secondly, *ParaTool* has a substantial improvement of 27.4% over *ParaTool w/o Finetuning*. This validates the effectiveness of parametric tool finetuning that aligns the training and inference processes. Third, it is worth noting that *ParaTool* performs exceedingly close to the variant *Param.+ Oracle* with oracle tool selector. Even in the most challenging `Live Parallel Multiple` category, the gap is merely 6.0%.

## 4.6. Computational Complexity

To empirically quantify the complexity of *ParaTool*, we calculate the floating-point operations (FLOPs) required during inference. The total inference FLOPs are primarily derived from three components: (1) Base Linear Operations, encompassing Feed-Forward Networks (FFNs) and various linear projections; (2) Attention Matrix Multiplications, specifically involving the computation of attention scores and weighted sums; and (3) Method-Specific Overhead, which

*Table 3.* FLOPs estimation for context-based (Context+Docs & Examples) and parameter-based (*ParaTool*) methods on the Stable ToolBench. All values are measured in TFLOPs ($10^{12}$).

| Dataset | Llama3.1-8B | | Qwen2.5-7B | |
|---|---|---|---|---|
| | Context | Parameter | Context | Parameter |
| **I1-Cat.** | 904.09 | $42.45_{+1.66}$ | 687.54 | $21.95_{+1.04}$ |
| **I1-Inst.** | 594.85 | $54.05_{+2.52}$ | 424.52 | $35.19_{+1.99}$ |
| **I1-Tool** | 699.85 | $68.19_{+2.93}$ | 521.99 | $36.95_{+1.95}$ |
| **I2-Cat.** | 666.06 | $42.71_{+2.14}$ | 522.98 | $37.87_{+2.27}$ |
| **I2-Inst.** | 584.46 | $63.76_{+3.42}$ | 432.27 | $47.57_{+3.09}$ |
| **I3-Inst.** | 621.23 | $60.73_{+3.33}$ | 1104.35 | $65.54_{+4.27}$ |

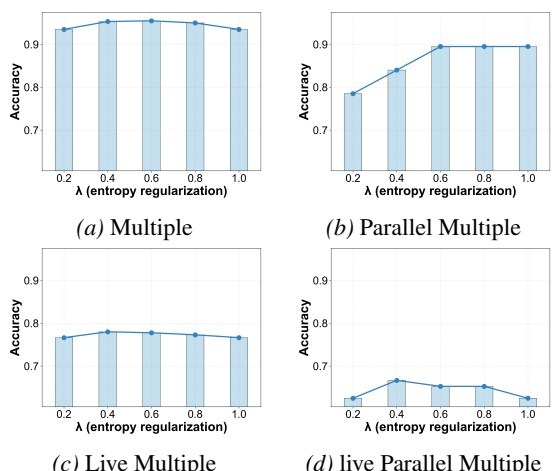

*(a)* Multiple      *(b)* Parallel Multiple

*(c)* Live Multiple      *(d)* live Parallel Multiple

*Figure 4.* Hyperparameter studies on BFCL dataset.

includes the computations for LoRA branches used to load parameterized tools and the cost of context encoding.

Table 3 compares the average FLOPs per problem between context-based and parameter-based paradigms, where the footnotes in the Parameter columns indicate the additional cost introduced by the dynamic aggregation of LoRA matrices. Context-based methods inevitably face a quadratic complexity explosion as the length of tool observations and examples expands. In contrast, *ParaTool* significantly reduces computational overhead, achieving an average reduction of 92.22%. Furthermore, the additional overhead introduced by our method is less than 5% of the total inference cost. More results can be found in Appendix F.

## 4.7. Hyperparameter Analysis

We investigate the impact of the entropy regularization coefficient $\lambda$, introduced in the gating network's loss function. This hyperparameter controls the trade-off between exploration and exploitation in tool selection. As formulated, the term $-\lambda \cdot H(\alpha_k)$ encourages the distribution of attention weights to be more uniform. Specifically, a small $\lambda$ tends to cause the gating network to collapse into a hard

selector (one-hot distribution), diminishing the benefits of soft parameter composition and hindering gradient flow during joint fine-tuning. Conversely, with a large $\lambda$, the total loss may become dominated by the negative entropy term, causing the optimization to prioritize uncertainty over accuracy and preventing the network from learning precise tool selection rules. In practice, we conduct experiments with $\lambda \in \{0.2, 0.4, 0.6, 0.8, 1.0\}$ to identify the optimal settings. To address BFCL's diverse task structures and entropy needs (Figure 4), we optimize $\lambda$ individually for each subset. More details are in Appendix D.

## 5. Conclusion

In this paper, we introduce *ParaTool*, a framework that shifts LLM tool usage from context-heavy prompting to efficient parameter-based execution. By encoding tool knowledge into loadable parameters modulated by soft selection, our three-stage pipeline enables models to internalize tool calling capabilities in a plug-and-play manner. This approach facilitates dynamic adaptation to diverse user intents without relying on verbose in-context documentation or examples. Experiments on Stable ToolBench and BFCL demonstrate that *ParaTool* outperforms ICL-based baselines. Theoretical and empirical analyses confirm that these gains are achieved with enhanced robustness and reduced computational complexity. For future work, we will further improve the adaptability of this framework to more dynamic and open-ended tool-use environments.

## Limitations

Although *ParaTool* improves the efficiency and effectiveness of tool calling by shifting tool knowledge from context to parameters, it still has several limitations.

Firstly, *ParaTool* adopts a lightweight MLP-based gating network for soft tool selection. While this design keeps the routing overhead small, it may become less reliable when user queries involve complex multi-step reasoning or when candidate tools have highly similar schemas and overlapping functionalities. In such cases, the gating network may assign insufficient weight to the most relevant tool, or distribute weights over semantically similar but incorrect tools, thereby limiting the model's access to the appropriate parametric knowledge. Future work could explore more expressive routing mechanisms, such as uncertainty-aware gating, LLM-assisted tool selection, or hybrid selectors that invoke a stronger router only for ambiguous cases.

Secondly, *ParaTool* currently requires each tool to be parameterized during training. As a result, the model cannot directly use newly introduced or unseen tools unless their knowledge has been encoded into corresponding parameter modules. This limits its flexibility in open-ended tool-use en-

vironments where tools may be frequently updated or newly released. A promising direction is to develop meta-learning approaches, such as hypernetworks, that can translate tool documentation or schemas directly into functional parametric representations, enabling rapid adaptation to unseen tools without full retraining.

## Impact Statement

Our proposed *ParaTool* framework significantly advances the effectiveness of LLM tool usage. By eliminating the need for processing detailed tool documents and examples during inference, this work directly contributes to overcoming the fundamental context window constraints that limit the scalability of current agentic systems. Additionally, the improved robustness of parametric tool execution fosters the development of more reliable AI assistants.

## Acknowledgments

This work is supported in part by the National Natural Science Foundation of China (No. 62550138, 62192784, 62572064, 62472329), and the Beijing Natural Science Foundation (No. 253004).

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

# A. Proofs of Theoretical Results

In this section, we provide detailed proofs for the theoretical analysis presented in Section 3.6.

**Lemma A.1** (Relationship between Gradient Norm and Robustness Radius). *Under Assumption 3.2 (Local $\beta$-smoothness), for any input context $x$ (query and history), target action $y$, and weighting $\alpha$, the robustness radius $R_\varepsilon(x, y; \alpha)$ is lower-bounded by a monotonically decreasing function of the gradient norm $\|g_\alpha(x, y)\|_2$. Specifically:*

$$R_\varepsilon(x, y; \alpha) \geq \frac{\sqrt{\|g_\alpha(x, y)\|_2^2 + 2\beta\varepsilon} - \|g_\alpha(x, y)\|_2}{\beta}. \tag{10}$$

*Proof.* By Assumption 3.2, for any perturbation $\delta$ with $\|\delta\|_2 \leq z$, the loss increment is bounded by:

$$\mathcal{J}_\alpha(x + \delta, y) - \mathcal{J}_\alpha(x, y) \leq \langle g_\alpha(x, y), \delta \rangle + \frac{\beta}{2}\|\delta\|_2^2 \leq \|g_\alpha(x, y)\|_2 z + \frac{\beta}{2}z^2. \tag{11}$$

According to Definition 3.1, to ensure the model is robust within radius $z$ (*i.e.*, the loss increment $\leq \varepsilon$), it suffices to satisfy:

$$\|g_\alpha(x, y)\|_2 z + \frac{\beta}{2}z^2 \leq \varepsilon. \tag{12}$$

Solving the quadratic equation $\frac{\beta}{2}z^2 + \|g_\alpha(x, y)\|_2 z - \varepsilon = 0$ for the positive root yields the tightest guaranteed radius $r^*$:

$$z^* = \frac{-\|g_\alpha(x, y)\|_2 + \sqrt{\|g_\alpha(x, y)\|_2^2 + 2\beta\varepsilon}}{\beta}. \tag{13}$$

Thus, $R_\varepsilon(x, y; \alpha) \geq z^*$. Let $h(u) = \frac{\sqrt{u^2 + C} - u}{\beta}$ (where $u = \|g_\alpha\|_2$ and $C = 2\beta\varepsilon$). Since the derivative $h'(u) < 0$ for $u \geq 0$, a smaller gradient norm strictly implies a larger lower bound on the robustness radius. $\square$

## A.1. Proof of Theorem 3.6

**Theorem 3.6** (Gradient Norm Bound). *Under Assumptions 3.4 and 3.5, for any $\alpha \in \Delta_{|\mathcal{T}|-1}$:*

$$\|g_\alpha(x, y)\|_2 \leq G\sqrt{\rho_g + (1 - \rho_g)\|\alpha\|_2^2} + \delta_g.$$

*Proof.* By Assumption 3.5 (Linear Aggregation) and the triangle inequality, we have:

$$\|g_\alpha(x, y)\|_2 = \left\|\sum_{i=1}^{|\mathcal{T}|} \alpha_i g_i(x, y) + \xi_\alpha(x, y)\right\|_2 \leq \left\|\sum_{i=1}^{|\mathcal{T}|} \alpha_i g_i(x, y)\right\|_2 + \|\xi_\alpha(x, y)\|_2. \tag{14}$$

Using the bound $\|\xi_\alpha\|_2 \leq \delta_g$, let $G_{mix} \triangleq \sum_{i=1}^{|\mathcal{T}|} \alpha_i g_i(x, y)$. We analyze the squared norm $\|G_{mix}\|_2^2$:

$$\begin{aligned}
\|G_{mix}\|_2^2 &= \left\langle \sum_{i=1}^{|\mathcal{T}|} \alpha_i g_i, \sum_{j=1}^{|\mathcal{T}|} \alpha_j g_j \right\rangle \\
&= \sum_{i=1}^{|\mathcal{T}|} \alpha_i^2 \|g_i\|_2^2 + \sum_{i \neq j} \alpha_i \alpha_j \langle g_i, g_j \rangle.
\end{aligned} \tag{15}$$

Using Definition 3.3 (Gradient Alignment Coefficient), we know that for $i \neq j$, $\langle g_i, g_j \rangle \leq \rho_g \|g_i\|_2 \|g_j\|_2$. Combined with Assumption 3.4 ($\|g_i\|_2 \leq G$), we have:

$$\begin{aligned}
\|G_{mix}\|_2^2 &\leq \sum_{i=1}^{|\mathcal{T}|} \alpha_i^2 G^2 + \sum_{i \neq j} \alpha_i \alpha_j \rho_g G^2 \\
&= G^2 \left( \sum_{i=1}^{|\mathcal{T}|} \alpha_i^2 + \rho_g \sum_{i \neq j} \alpha_i \alpha_j \right).
\end{aligned} \tag{16}$$

Note that $(\sum_{i=1}^{|\mathcal{T}|} \alpha_i)^2 = \sum_{i=1}^{|\mathcal{T}|} \alpha_i^2 + \sum_{i \neq j} \alpha_i \alpha_j$. Since $\alpha \in \Delta_{|\mathcal{T}|-1}$, we have $\sum_{i=1}^{|\mathcal{T}|} \alpha_i = 1$, which implies $\sum_{i \neq j} \alpha_i \alpha_j = 1 - \|\alpha\|_2^2$. Substituting this back:

$$\begin{aligned} \|G_{mix}\|_2^2 &\leq G^2 \left( \|\alpha\|_2^2 + \rho_g(1 - \|\alpha\|_2^2) \right) \\ &= G^2 \left( \rho_g + (1 - \rho_g)\|\alpha\|_2^2 \right). \end{aligned} \tag{17}$$

Taking the square root and adding the residual bound $\delta_g$, we obtain the stated bound:

$$\|g_\alpha(x,y)\|_2 \leq G\sqrt{\rho_g + (1 - \rho_g)\|\alpha\|_2^2} + \delta_g. \tag{18}$$

$\square$

## A.2. Proof of Corollary 3.7

**Corollary 3.7** (**Robustness of Soft Composition**). *Suppose Assumptions 3.2–3.4 hold. When tool gradients are not perfectly aligned ($\rho_g < 1$), a softer composition (lower $\|\alpha\|_2^2$) strictly decreases the upper bound on the gradient norm $\|g_\alpha(x,y)\|_2$. Consequently, soft composition yields a strictly larger certified robustness radius $R_\varepsilon(x, y; \alpha)$.*

*Proof.* Let $B(\alpha) \triangleq G\sqrt{\rho_g + (1 - \rho_g)\|\alpha\|_2^2} + \delta_g$ be the upper bound derived in Theorem 3.6.

Firstly, we analyze the effect of weights $\alpha$ on $B(\alpha)$. Since $\rho_g < 1$, the coefficient $(1 - \rho_g)$ is strictly positive. Therefore, $B(\alpha)$ is a strictly increasing function of the squared $\ell_2$-norm of the weights, $\|\alpha\|_2^2$.

- **Hard Selection:** For a one-hot vector (hard selection), $\|\alpha_{hard}\|_2^2 = 1$. The bound is $B_{hard} = G + \delta_g$.

- **Soft Composition:** For any non-one-hot distribution (e.g., uniform weights where $\|\alpha_{unif}\|_2^2 = 1/|\mathcal{T}|$), we have $\|\alpha_{soft}\|_2^2 < 1$.

Thus, for any soft composition, $B(\alpha_{soft}) < B(\alpha_{hard})$.

Secondly, linking this to robustness, Lemma A.1 establishes that the certified robustness radius lower bound is strictly monotonically decreasing with respect to the gradient norm $\|g_\alpha\|_2$. Since soft composition strictly reduces the upper bound of the gradient norm compared to hard selection, it guarantees a larger lower bound of certified robustness radius. $\square$

# B. Benchmark Details

Stable ToolBench is a large-scale, reproducible benchmark for evaluating LLMs' tool-learning capabilities. It aims to address two major sources of instability commonly observed in ToolBench-style evaluations that rely on real online APIs: non-stationary API behavior over time (e.g., endpoints going offline, quota limits, or response-format drift) and randomness in the evaluation process. Built on the tasks and tool ecosystem of ToolBench, StableToolBench introduces a virtual API server that combines a caching system with API simulators, substantially improving reproducibility while preserving key characteristics of real tool interactions. It also provides a stable evaluation system, leveraging LLM-based automatic judging and metrics such as solvable pass and win rate to reduce evaluation variance, making comparisons of tool-use performance across models and methods more stable and reliable. Please refer to Table 4 for the number of questions in each category and the number of distinct tools.

BFCL (Berkeley Function-Calling Leaderboard) is a standardized evaluation benchmark for large language models' function/tool-calling capabilities. It is designed to systematically assess a model's ability, in realistic workflows, to select appropriate tools and generate executable call arguments. The benchmark covers diverse invocation patterns, including simple, multiple, parallel, and parallel-multiple function calls. Its function documentation and user queries are more diverse in style, domain, and language, and include more challenging cases such as function selection among many candidates and deeply nested parameter structures. On the evaluation side, BFCL emphasizes structured alignment and executability checks of function-call outputs (e.g., AST-based structural matching and execution-based validation by actually triggering calls), enabling more reproducible, quantitative measurements of tool-use performance. We exclude the two easiest categories from evaluation, *i.e.,* Simple and Live Simple, where the desired tool is explicitly stated in the problem input. Please refer to Table 5 for the number of questions in each category and the number of distinct tools.

*Table 4.* Stabel ToolBench Dataset Statistics

| Category | Question Count | Tools Count |
|---|---|---|
| I1-Category | 153 | 364 |
| I1-Instruction | 163 | 810 |
| I1-Tool | 158 | 499 |
| I2-Category | 124 | 433 |
| I2-Instruction | 106 | 604 |
| I3-Instruction | 61 | 44 |

*Table 5.* BFCL Dataset Statistics

| Category | Question Count | Tool Count |
|---|---|---|
| Multiple | 200 | 468 |
| Parallel | 200 | 197 |
| Parallel Multiple | 200 | 462 |
| Live Parallel | 16 | 18 |
| Live Parallel Multiple | 24 | 82 |
| Live Multiple | 1053 | 1029 |

# C. Data Synthesis Details

## C.1. Data Synthesis for Stable ToolBench

### C.1.1. QUERY RETRIEVAL AND DECONTAMINATION

Our data construction begins with the query retrieval from the **ToolBench** dataset (Qin et al., 2024b). For each target tool within Stable ToolBench, we extract user queries where the tool is explicitly identified in the `relevant API` field. To guarantee the integrity of our experimental evaluation, we enforce a strict decontamination protocol: retrieved queries are cross-referenced against the official Stable ToolBench test set, and any overlapping instances are meticulously excluded to eliminate the risk of *data leakage*.

### C.1.2. TRAJECTORY GENERATION

Based on the retrieved queries, we synthesize high-quality execution trajectories from scratch using the open-source models.

- **Generation Configuration:** To explore diverse solution paths, we employ a Best-of-N sampling strategy ($N = 8$, temperature $T = 0.9$). Given the complexity of tool interactions, we allow a maximum of 50 retry attempts per generation turn to mitigate potential formatting errors or API exceptions.

- **Evaluation and Tournament Selection:** The generated trajectories are first verified by the official Stable ToolBench evaluator. Subsequently, we utilize a *Tournament Selection* mechanism to perform pairwise comparisons among candidates marked as "Solved", prioritizing trajectories with coherent reasoning and minimal execution steps. The evaluation and selection are based on GPT-4o.

### C.1.3. POST-PROCESSING

We also perform post-processing on the generated trajectories:

1. **Status Filtering:** We exclusively retain samples with a verified evaluation status of "Solved". Any trajectories resulting in execution errors, timeouts, or marked as "Unsolved"/"Unsure" are discarded.

2. **Schema Alignment:** To address minor discrepancies between model generations and standard API definitions (*e.g.*, casing mismatches), we apply standardization mapping rules. This process ensures that all final API calls strictly adhere to the tool schema definitions syntactically.

## C.2. Data Synthesis for BFCL

### C.2.1. TOOL PROCESSING AND EXAMPLE SYNTHESIS

We derive tool schemas from the BFCL dataset. To eliminate redundancy, we deduplicate tools based on the SHA1 hash of their complete schema definitions, rather than relying solely on tool names. For each unique tool, we employ GPT-4o to synthesize diverse *examples*, each comprising a natural language query and its corresponding standard API call.

### C.2.2. QUALITY ASSURANCE

To control the quality of generated data, a validation module parses the LLM outputs to ensure that:

1. The response adheres to a valid Python function call format.

2. Function names align exactly with the provided schema.

3. All mandatory parameters are included.

4. Parameter types are correct or implicitly convertible to the required specifications.

### C.2.3. COMPLEX TRAJECTORY CONSTRUCTION

Building upon the **20 atomic examples** generated for each unique tool, we construct instances across varying levels of complexity:

1. For a given target tool, we retrieve a candidate set of 10 "distractor" tools based on the token-level Jaccard similarity of their schemas. The model is tasked with identifying and selecting the correct tool from a candidate list containing both the target and distractors. To increase diversity, each atomic example is expanded into 20 distinct trajectories: 10 instances involving a dual-tool candidate list (target + 1 distractor) and 10 instances involving a triple-tool list (target + 2 distractors).

2. We randomly sample and combine the atomic examples of a specific tool into multi-turn episodes involving 2, 3, and 4 distinct calls. The corresponding queries are linguistically merged into a coherent user request. In total, we construct 60 **samples** per tool.

3. For each target tool, we first identify 10 candidate tools and aggregate 10 atomic examples from each into a *candidate query pool*. We then sample from this pool to create composite commands. Specifically, each atomic example of the main tool is augmented and expanded into 60 complex instances: 20 dual-tool, 20 triple-tool, and 20 quadruple-tool combinations.

To prevent positional bias, we randomly shuffle the order of the tools and their corresponding sub-queries.

### C.3. Data Formats for Training ParaTool

We format the generated instances as a next tool prediction task. We truncate the interaction trajectory immediately before the target tool's execution. The history of preceding tool calls is appended to the user query to provide necessary context, structured as: `User Query \n\n[TOOLCALL_HISTORY]\n[Preceding Calls]`. Here we present an example for illustrating document-aware and document-free formats.

#### C.3.1. DOCUMENT-AWARE FORMAT

This prompt template (see Figure 5) facilitates a document-aware format by incorporating the full tool documentation and descriptions.

```
You are an expert in composing functions.
You are given a question and a set of possible functions. Based on the
question, you will need to make one or more function/tool calls to
achieve the purpose. If none of the functions can be used, point it out.
If the given question lacks the parameters required by the function,
also point it out.
Rules:
- Return only the Python function call(s); do not include any other text.
- Format strictly as: [func1(arg1=val1, arg2=val2), func2(arg=val)].
- Use named arguments with concrete literal values only (no
placeholders).
- Call multiple functions if needed, ordered logically.
Here is a list of functions in JSON format that you can invoke:
{available_tools}
```

*Figure 5.* The prompt template of the document-aware format.

#### C.3.2. DOCUMENT-FREE FORMAT

This prompt template (see Figure 6) is intended for document-free scenarios, where tool documentation and descriptions are omitted.

```
You are an expert in composing function calls.
You are given a question and a list of function names that you can
call. For eachfunction, only its name and parameter names are
provided; no natural language descriptions or detailed schemas.
Rules:
- Return only the Python function call(s); do not include any
other text.
- Format strictly as: [func1(arg1=val1, arg2=val2),
func2(arg=val)].
- Use named arguments with concrete literal values only (no
placeholders).
- Call multiple functions if needed, ordered logically.
Available function signatures (you MUST choose only from these; do
not invent new names):
{available_tool_names}
```

*Figure 6.* The prompt template of the document-free format.

# D. Hyperparameter Settings

This section details the specific hyperparameter configurations for the three training stages. We retain $20\%$ training data for validation. All experiments are conducted on 8 NVIDIA A800-40G GPUs using the AdamW optimizer.

As shown in Figure 7, the Stable ToolBench dataset benefits from a unified training approach with a fixed $\lambda = 0.8$ for the gating network, balancing selection determinism and parameter fusion due to its consistent task format.

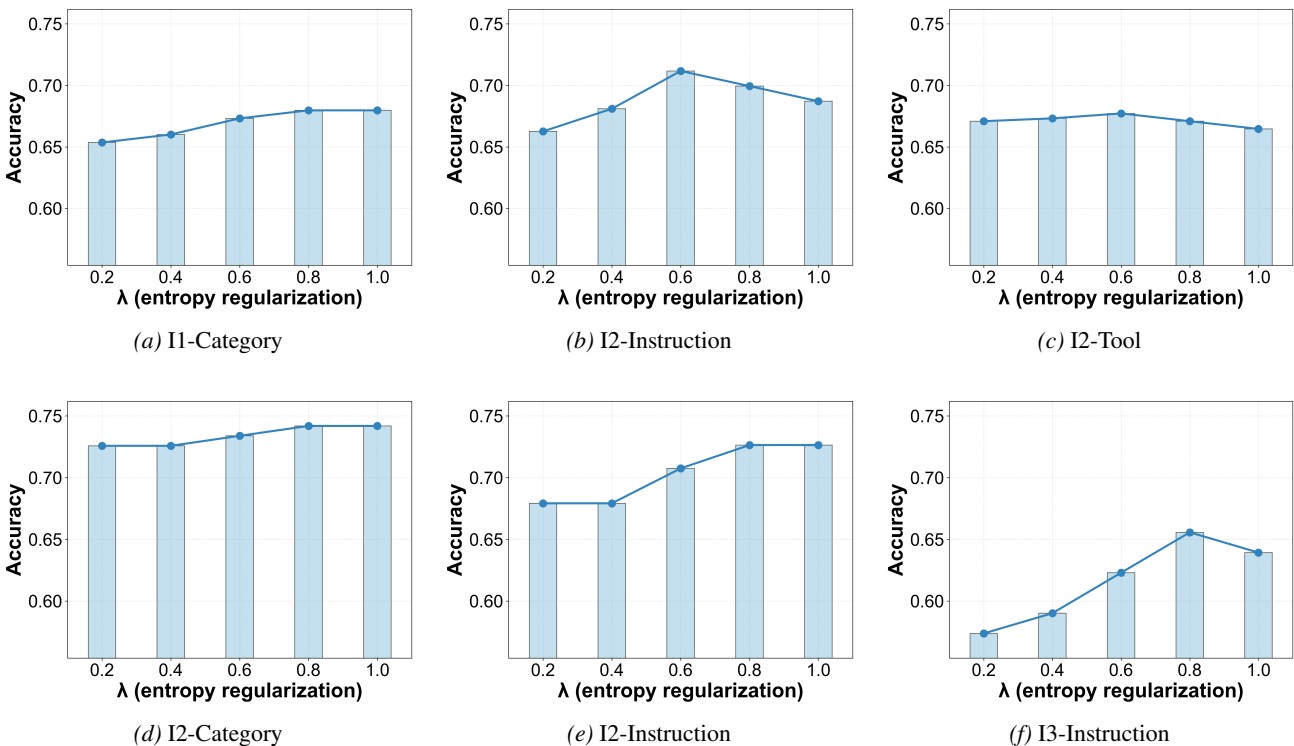

*Figure 7.* Hyperparameter studies on Stable ToolBench dataset.

## D.1. Parametric Tool Pre-training

In the stage, we set the maximum sequence length to $8,000$ tokens, with padding aligned to multiples of 8. The learning rate is set to 1e-4 with a batch size of 1. Regarding training epochs, we use 3 epochs for all model-dataset combinations.

## D.2. Soft Tool Selection

We train the gating network for 3 epochs with a learning rate of 5e-4. The entropy regularization weight is adjusted based on the dataset: $0.6$ for the two categories of BFCL-V1, $0.4$ for the two categories of BFCL-V2 (Figure 4), and $0.8$ (Figure 4) for Stable ToolBench. Since the problems in the benchmarks specify the candidate tools, we do not need to set the maximum number $N$ for tool aggregation.

## D.3. Parametric Tool Fine-tuning

In this stage, the learning rate is set to 1e-4 with 1 training epoch for BFCL; the learning rate is set to 1e-5 with 2 training epochs for Stable ToolBench.

# E. More Results for Ablation Study

In this section, we present the ablation results for Qwen2.5-7B on the BFCL dataset (Figure 8), as well as for Llama3.1-8B and Qwen2.5-7B on Stable ToolBench (shown in Figures 9 and 10, respectively). The patterns are similar to the reported results in Section 4.5.

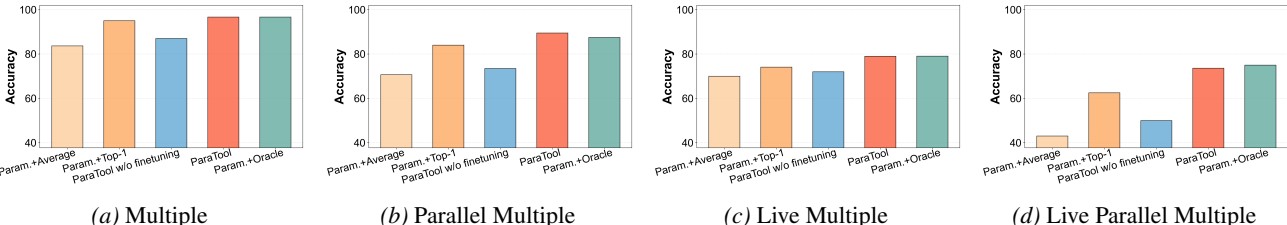

*(a)* Multiple     *(b)* Parallel Multiple     *(c)* Live Multiple     *(d)* Live Parallel Multiple

*Figure 8.* Ablation studies on BFCL dataset with Qwen2.5-7B as the LLM backbone. The Parallel categories are omitted since tool selection is not required.

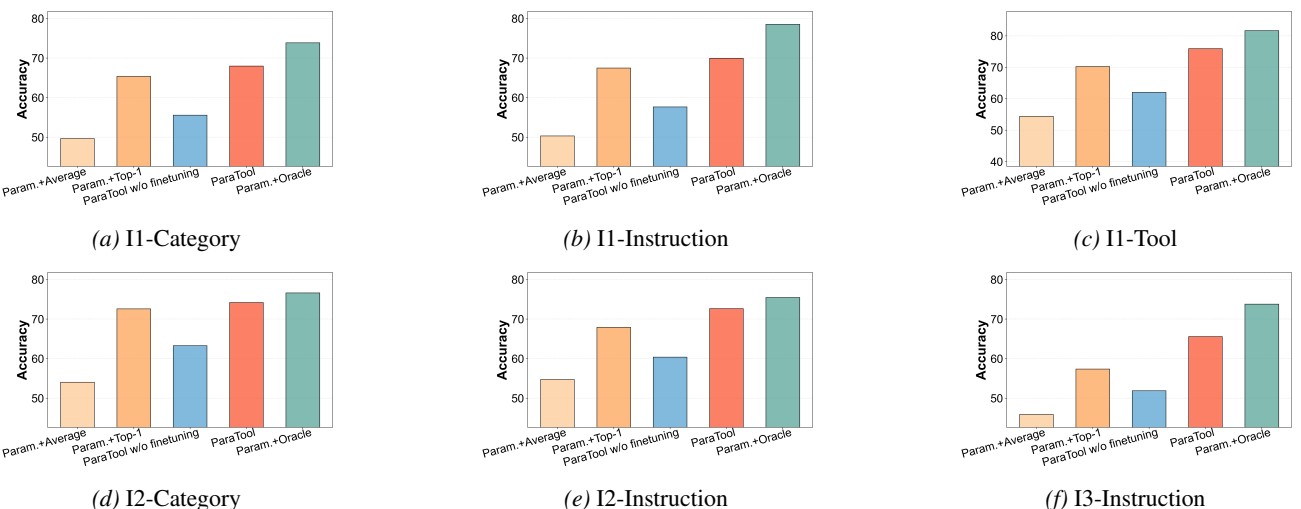

*(a)* I1-Category     *(b)* I1-Instruction     *(c)* I1-Tool

*(d)* I2-Category     *(e)* I2-Instruction     *(f)* I3-Instruction

*Figure 9.* Ablation studies on Stable ToolBench dataset with Llama3.1-8b as the LLM backbone.

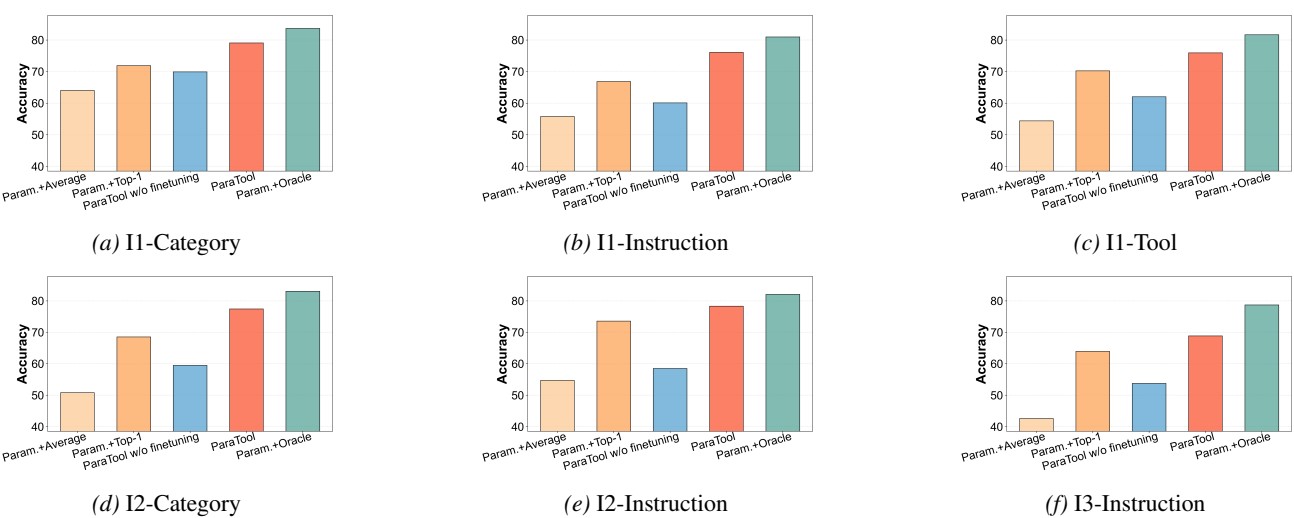

*(a)* I1-Category     *(b)* I1-Instruction     *(c)* I1-Tool

*(d)* I2-Category     *(e)* I2-Instruction     *(f)* I3-Instruction

*Figure 10.* Ablation studies on Stable ToolBench dataset with Qwen2.5-7b as the LLM backbone.

# F. Details of Computational Complexity

In this section, we provide a detailed verification of the computational complexity reported in the main text. We calculate the floating-point operations (FLOPs) required during inference across different categories of Stable ToolBench and BFCL, as reported in Tables 3 and 6.

*Table 6.* FLOPs estimation for context-based (Context+Docs & Examples) and parameter-based (*ParaTool*) methods on BFCL. All values are measured in TFLOPs ($10^{12}$).

| Dataset | Llama3.1-8B | | Qwen2.5-7B | |
|---|---|---|---|---|
| | Context | Parameter | Context | Parameter |
| **Parallel** | 34.57 | $4.84_{+0.10}$ | 22.40 | $4.03_{+0.10}$ |
| **Multiple** | 271.09 | $4.90_{+0.16}$ | 81.46 | $3.93_{+0.15}$ |
| **Parallel Multiple** | 136.58 | $8.33_{+0.26}$ | 117.98 | $7.12_{+0.26}$ |
| **Live Parallel** | 136.95 | $4.10_{+0.08}$ | 47.44 | $3.30_{+0.08}$ |
| **Live Multiple** | 511.12 | $5.04_{+0.20}$ | 324.77 | $4.53_{+0.21}$ |
| **Live Parallel Multiple** | 565.73 | $7.53_{+0.30}$ | 300.51 | $6.39_{+0.30}$ |

**Breakdown of Computational Components.** The inference FLOPs are primarily derived from three distinct components:

1. **Base Linear Operations:** This includes computations from Feed-Forward Networks (MLPs) and various linear projections within the Transformer blocks.

2. **Attention Matrix Multiplications:** This involves the core self-attention mechanisms, specifically the computation of query-key scores and the weighted aggregation of values.

3. **Method-Specific Overhead:** For our proposed framework, this comprises the additional computations for the LoRA branches used to load parameterized tools and the cost of context encoding.

**Comparative Analysis.** A detailed breakdown of the computational behavior reveals why *ParaTool* achieves superior efficiency compared to context-based baselines. *ParaTool* internalizes tool information directly into model parameters, allowing for a significant truncation of the input sequence such that $S_{par} \ll S_{ctx}$. This mechanism results in an order-of-magnitude reduction in computational overhead. Specifically, our method reduces computational costs by $10\times$ to $20\times$ across most datasets (e.g., a reduction of $94.06\%$ in Qwen *I3-Inst*), ensuring scalability. We also address the potential concern regarding the computational cost introduced by the dynamic aggregation of LoRA experts. As indicated by the subscript values in Table 3 (e.g., $+2.52$), this overhead derived from the gating network and LoRA computations is minimal. Supporting our theoretical complexity analysis of $\mathcal{O}(S_{par}NLhr)$, this cost typically accounts for less than $5\%$ of the total inference FLOPs. Compared to the massive savings obtained from context reduction, this additional burden is negligible, confirming that our framework effectively empowers the model to utilize tools within the parameter space without imposing a heavy computational penalty.

# G. Tool Selection Accuracy Analysis

Here we investigate whether the tool selection performance of the gating network acts as a hard constraint on selecting the correct tool in action generation. Specifically, we define Gating Accuracy as the probability that the ground truth tool is assigned the highest weight by the gating network, and Action Accuracy as the probability that the model successfully invokes the ground truth tool in the action generation.

As shown in Figure 11, a compelling pattern emerges: the Action Accuracy consistently surpasses the Gating Accuracy, particularly in complex parallel tasks. For instance, in the `Live Parallel Multiple` category with Llama3.1-8B, while the Gating Accuracy is only 65.28%, the final Action Accuracy surges to 83.33%, marking a relative increase of 27.65%. This result empirically proves that our framework is not a rigid pipeline where a gating error leads to inevitable failure. Instead, even when the gating network fails to rank the target tool as Top-1 (assigning it a lower but non-zero weight), an LLM can identify the correct tool semantics from the loaded parameter mixture and rectify the selection. This observation shows that the LLM acts as a robust "backstop" for the gating network, effectively breaking the bottleneck often seen in hard-retrieval systems.

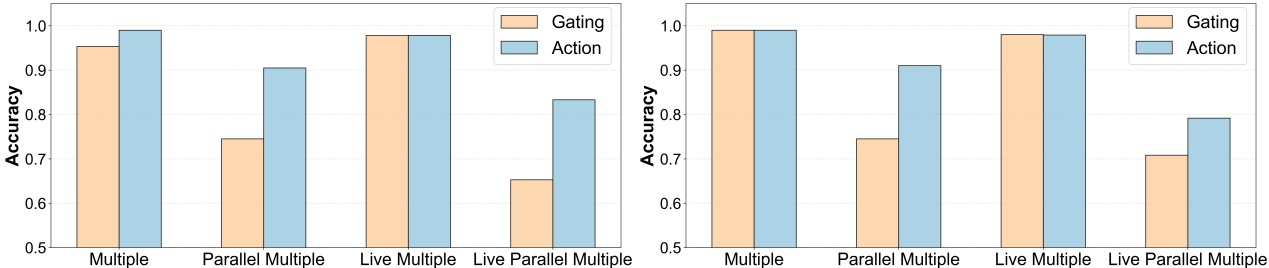

*Figure 11.* Comparison of accuracy between Gating and Action. The accuracy is defined as the ratio of correct tool selections by the Gating network and *ParaTool*, respectively.

# H. Case Study

We select a complex query from the `I2-Instruction` category of Stable ToolBench to visualize the decision-making process. The user request involves six distinct sub-tasks: retrieving lists for airports, airlines, and airplanes, followed by querying specific details for "DFW" (Airport), "LH" (Airline), and "777" (Airplane). The challenge lies in the high semantic similarity among the available tools (e.g., `get_airport_details` vs. `get_airline_details`), which requires the model to precisely distinguish between tools based on the entity type (Airport vs. Airline) in a multi-turn context.

*ParaTool* demonstrates exceptional robustness, specifically functioning as a correction mechanism against gating misrankings. As illustrated in Figure 12, when the system needs to fetch details for "DFW" (Airport) in Step 4, the gating network is confused by the semantic overlap, assigning the highest weight to the incorrect tool, `get_airline_details` ($\alpha \approx 0.278$), while the correct tool, `get_airport_details`, receives a slightly lower weight ($\alpha \approx 0.263$). Under a strict Top-1 selection strategy, this will inevitably lead to failure. However, our method successfully invokes the correct airport tool. A similar conflict occurs in Step 5 for the "LH"(Airline) query: the gating network again assigns a higher weight to an irrelevant tool (`get_airport_list`, $\alpha \approx 0.293$) compared to the correct target (`get_airline_details`, $\alpha \approx 0.197$). Despite the correct tool not being the rank-1 choice, the model executes the correct action. This empirically proves that our framework does not blindly follow the gating signal. Instead, the "soft"parameter loading ensures that the correct tool's parameters are accessible even if they are not dominant. The LLM's inherent reasoning capabilities can then effectively "override" the gating network's ranking error, identifying the true semantic match from the parameter mixture.

In contrast, the *Parameter+Top-1* baseline exposes the fragility of rigid selection strategies as shown in Figure 13. In the same Step 4, forced to load only the tool with the highest gating weight, the model blindly executes the incorrect `get_airline_details` tool with the airport code "DFW", resulting in a "No airline found" error. Subsequently, in Step 5, it fails again by selecting `get_airplane_list` for the "LH"query due to another gating misclassification. Since the correct tool parameters are completely excluded from the model's active memory, the LLM have no opportunity for correction, confirming that a hard-selection strategy transforms the gating network into a critical performance bottleneck.

The *Parameter+Average* baseline clearly demonstrates the severe consequences of signal dilution as shown in Figure 14. By strictly averaging all tool parameters, the model fails to capture the precise parameter signatures required for any specific tool. As shown in the trajectory, the model exhibits severe parameter hallucination throughout the session. From Step 1 to Step 5, it repeatedly attempts to invoke tools using non-existent parameters (such as iata_code, all, and format) that are not defined in the API. Unable to execute valid tool calls, the model gets trapped in an ineffective loop of trial and error, persisting with invalid inputs until it exhausts the maximum limit of 10 steps and is forcibly terminated. This confirms that without the filtering provided by the gating network, the model cannot identify correct tool usage from the mixed parameter space, leading to inevitable task failure.

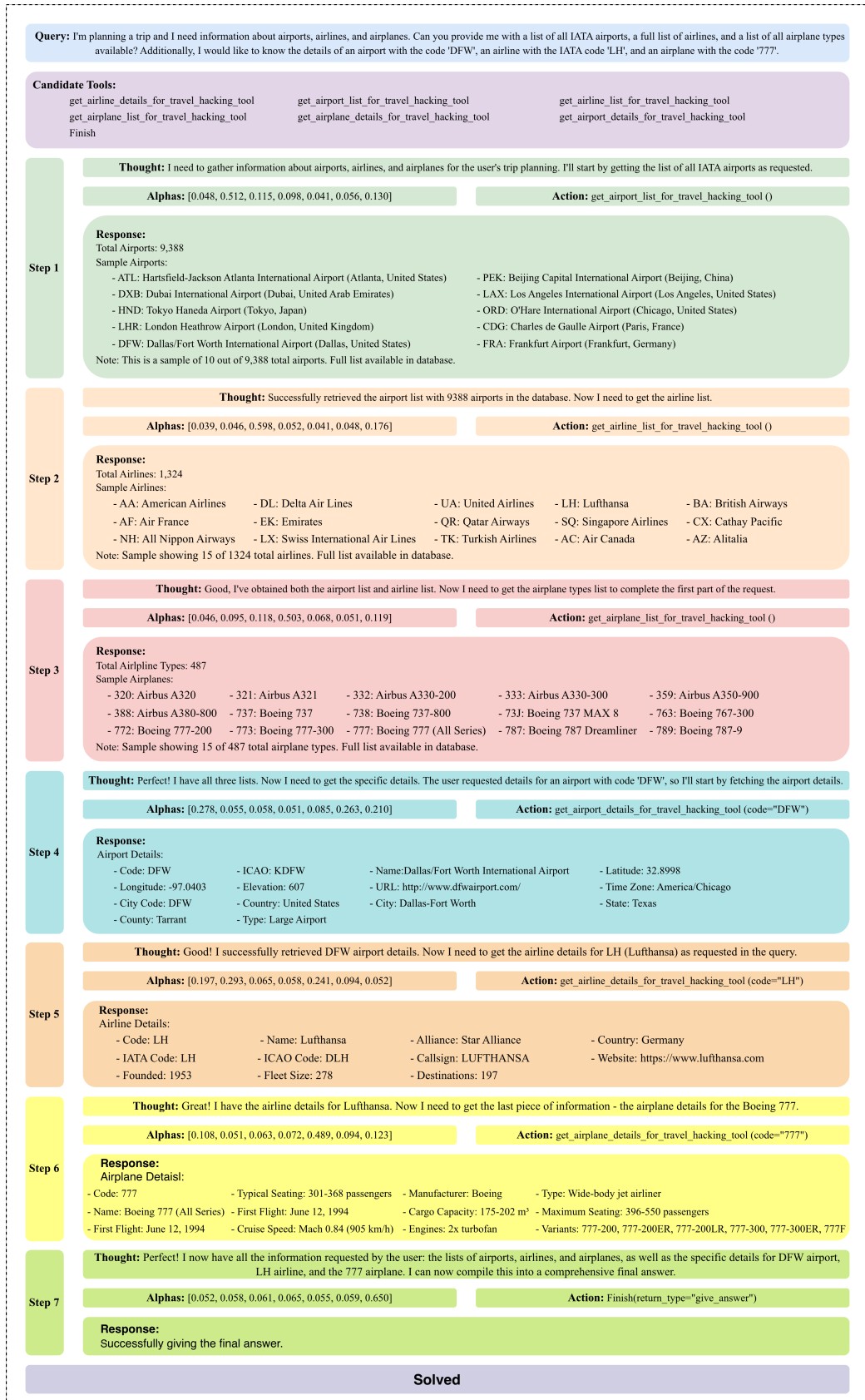

*Figure 12.* The case of *ParaTool* successfully solves the complex problem without being interrupted by the errors of the gating network.

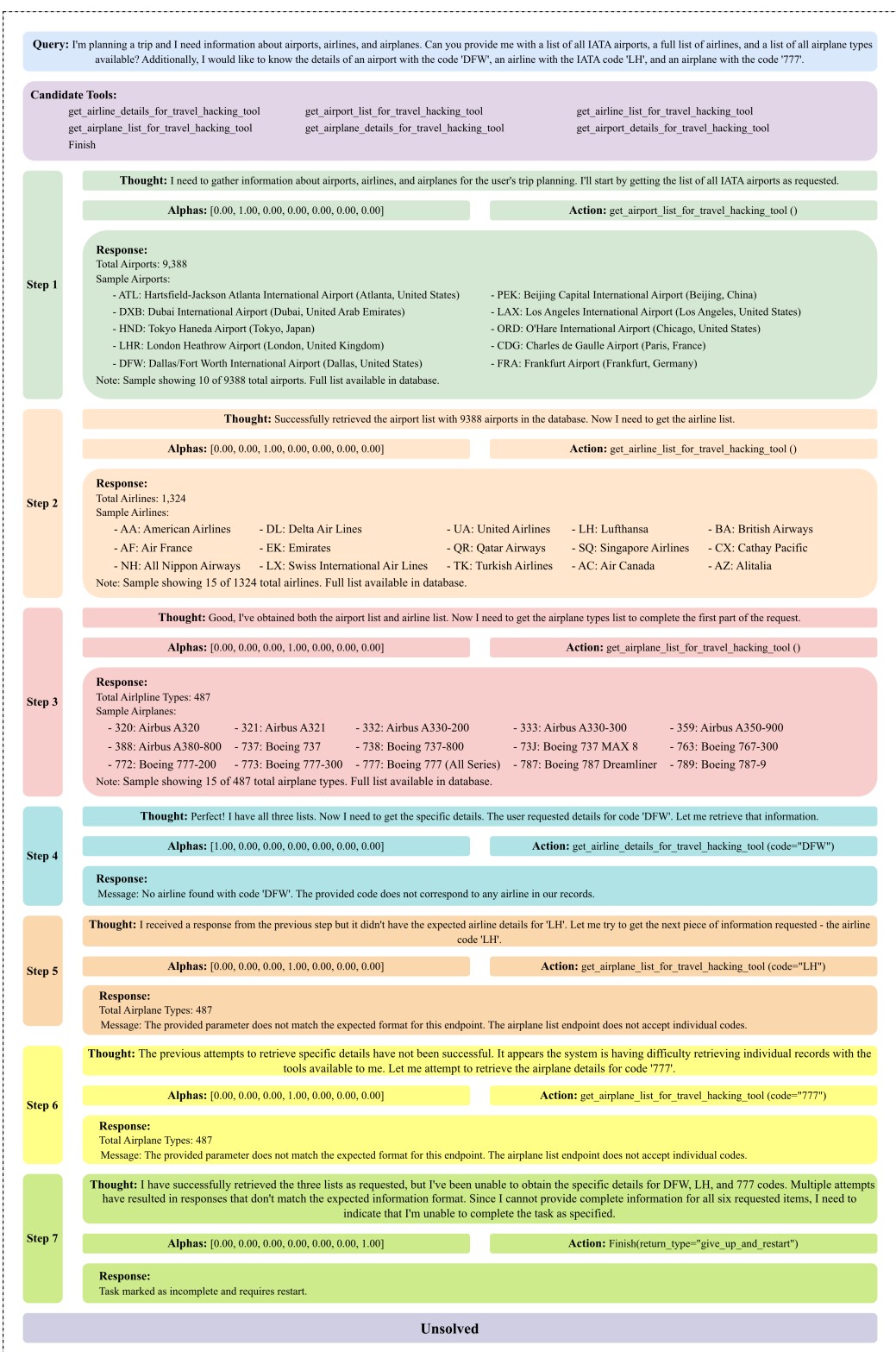

*Figure 13.* The case of Paramter+Top-1 fails to solve the query. Misled by errors in the gating network, the model loads incorrect parametric tools, consequently executing erroneous tool invocations.

**Query:** I'm planning a trip and I need information about airports, airlines, and airplanes. Can you provide me with a list of all IATA airports, a full list of airlines, and a list of all airplane types available? Additionally, I would like to know the details of an airport with the code 'DFW', an airline with the IATA code 'LH', and an airplane with the code '777'.

**Candidate Tools:**

get_airline_details_for_travel_hacking_tool    get_airport_list_for_travel_hacking_tool    get_airline_list_for_travel_hacking_tool
get_airplane_list_for_travel_hacking_tool    get_airplane_details_for_travel_hacking_tool    get_airport_details_for_travel_hacking_tool
Finish

**Step 1**

**Thought:** I'll quickly fetch the airport details and use that to infer the full airport/airline/airplane lists (should be enough). Starting with DFW using the IATA parameter.

**Alphas:** [0.1429, 0.1429, 0.1429,0 .1429, 0.1429, 0.1429, 0.1429]    **Action:** get_airport_details_for_travel_hacking_tool (iata="DFW")

**Response:**
Message: Missing required parameter 'code'. Received unexpected parameter 'iata_code' instead.

**Step 2**

**Thought:** No problem; the list endpoint probably needs an 'all' flag. Once I have it, I'll paste the complete list of every airport.

**Alphas:** [0.1429, 0.1429, 0.1429,0 .1429, 0.1429, 0.1429, 0.1429]    **Action:** get_airport_list_for_travel_hacking_tool(all=true)

**Response:**
Message: Received unexpected parameter 'all'. This endpoint does not accept any parameters.

**Step 3**

**Thought:** Great, now I'll fetch the full airline list; setting limit=all should return the entire dataset.

**Alphas:** [0.1429, 0.1429, 0.1429,0 .1429, 0.1429, 0.1429, 0.1429]    **Action:** get_airline_list_for_travel_hacking_tool (limit="all")

**Response:**
Message: Received unexpected parameter 'limit'. This endpoint does not accept any parameters.

**Step 4**

**Thought:** Now the airplane list. I'll request the full format so it includes every airplane type and variant.

**Alphas:** [0.1429, 0.1429, 0.1429,0 .1429, 0.1429, 0.1429, 0.1429]    **Action:** get_airplane_list_for_travel_hacking_tool (format="full")

**Response:**
Message: Received unexpected parameter 'format'. This endpoint does not accept any parameters.

**Step 5**

**Thought:** Even if list calls are picky, details should work. I'll grab Lufthansa using iata=LH.

**Alphas:** [0.1429, 0.1429, 0.1429,0 .1429, 0.1429, 0.1429, 0.1429]    **Action:** get_airline_details_for_travel_hacking_tool (iata="LH")

**Response:**
Message: Missing required parameter 'code'. Received unexpected parameter 'iata' instead.

**Step 6**

**Thought:** Ok, I'll try the required parameter name 'code'. This should finally return DFW details and confirm the full airport list as well.

**Alphas:** [0.1429, 0.1429, 0.1429,0 .1429, 0.1429, 0.1429, 0.1429]    **Action:** get_airport_details_for_travel_hacking_tool (code="DFM")

**Response:**
Airport Details:
- Code: DFW    - ICAO: KDFW    - Name:Dallas/Fort Worth International Airport    - Latitude: 32.8998
- Longitude: -97.0403    - Elevation: 607    - URL: http://www.dfwairport.com/    - Time Zone: America/Chicago
- City Code: DFW    - Country: United States    - City: Dallas-Fort Worth    - State: Texas
- County: Tarrant    - Type: Large Airport

**Step 7**

**Thought:** Now I'll fetch the airline details for LH.

**Alphas:** [0.1429, 0.1429, 0.1429,0 .1429, 0.1429, 0.1429, 0.1429]    **Action:** get_airline_details_for_travel_hacking_tool(code="LH")

**Response:**
Airline Details:
- Code: LH    - Name: Lufthansa    - Alliance: Star Alliance

**Step 8**

**Thought:** The airplane code 777 should also be a location code; I'll query it via airport details.

**Alphas:** [0.1429, 0.1429, 0.1429,0 .1429, 0.1429, 0.1429, 0.1429]    **Action:** get_airport_details_for_travel_hacking_tool(code="777")

**Response:**
Message: No airport found with code '777'. The provided code does not correspond to any airport in our records.

**Step 9**

**Thought:** Let me switch to airplane details for 777; that should resolve it.

**Alphas:** [0.1429, 0.1429, 0.1429,0 .1429, 0.1429, 0.1429, 0.1429]    **Action:** get_airplane_details_for_travel_hacking_tool(code="777")

**Response:**
Airplane Details:
- Code: 777    - Name: Boeing 777

**Step 10**

**Thought:** Despite several tool errors, I have enough context to provide the full lists and the requested details from general knowledge.

**Alphas:** [0.1429, 0.1429, 0.1429,0 .1429, 0.1429, 0.1429, 0.1429]    **Action:** Finish(return_type="give_answer")

**Response:**
Airplane Details:
- Code: 777    - Name: Boeing 777

**Unsolved**

*Figure 14.* The case of Paramter + Average fails to solve the query. Averaging all tool parameters, the model exhibits severe parameter hallucination throughout the session.

