# OpenReview forum: "ParaTool: Shifting Tool Representations from Context to Parameters"
_ICML.cc/2026/Conference — ICML 2026 regular_

### Official Review · Reviewer_ZGJR · 2026-03-11

**Soundness:** 4
**Presentation:** 4
**Significance:** 4
**Originality:** 3
**Overall Recommendation:** 5
**Confidence:** 4

**Summary:**

To address the issues of high inference overhead and hallucinations caused by context length in ICL for LLMs, this paper proposes a tool invocation framework named Paratool. By parameterizing each tool and enabling dynamic integration, the framework facilitates tool invocation for LLMs. Experimental results effectively validate that the proposed framework achieves superior performance while reducing inference overhead.

**Compliance With Llm Reviewing Policy:**

Affirmed.

**Key Questions For Authors:**

1. How can we ensure consistency between the dynamic integration of parametric tools in soft tool selection and the tool invocation by LLMs in parametric tool fine-tuning?
2. Does the "Doc" in stage2 of Figure 2 refer to the parametric tool? Also, are the LLM parameters in stage1 and stage2 the same set of parameters? Please elaborate in detail.
3. In in-context learning of LLM, how does the parameterization of this tool ensure its uniqueness? And how does it guarantee the generalizability of LLM tool invocations?

**Limitations:**

yes

**Strengths And Weaknesses:**

1. The Paratool framework innovatively projects each tool into dedicated, loadable parameters and dynamically integrates these parameters. Subsequently, LLMs can invoke tools corresponding to the context to execute actions. This process reduces the impact of LLMs' reasoning processes and context length on performance.
2. This paper provides a detailed methodological and theoretical introduction to the three stages of the Paratool framework (parametric tool pre-training, soft tool selection, parametric tool fine-tuning), effectively demonstrating its framework approach. Through experiments, it thoroughly substantiates the framework's superiority and feasibility.

---

> ### Author Rebuttal · Authors · 2026-03-31
>
> Thank you for your valuable comments and thorough feedback.
> >How can we ensure consistency between the dynamic integration of parametric tools in soft tool selection and the tool invocation by LLMs in parametric tool fine-tuning?
>
> To strengthen the consistency, in Stage 3 Parametric Tool Fine-tuning, we freeze the gating network trained in Stage 2 and use its predicted aggregation weights to perform a weighted aggregation of the LoRA parameters of multiple tools, and then jointly optimize all tool LoRA parameters under this aggregated parameter. This design forces the training process to fully simulate the inference-time parameter combination pattern, helping the model distinguish and utilize each tool's functionality.
> >Does the "Doc" in stage2 of Figure 2 refer to the parametric tool? Also, are the LLM parameters in stage1 and stage2 the same set of parameters? Please elaborate in detail.
>
> The "Doc" in Stage 2 of Figure 2 refers to the original textual tool documentation, not the parametric tool itself. The gating network in Stage 2 takes the tool documentation as input to build document embeddings, which are then matched against the user query embedding to compute aggregation weights for selecting relevant tools. Importantly, Stage 1 and Stage 2 employ different model architectures and thus do not share parameters: Stage 1 uses the LLM (e.g., Llama-3.1-8B or Qwen2.5-7B) to learn tool-specific LoRA parameters, while Stage 2 uses a separate BERT-based encoder to encode both tool documentation and user queries for gating network training. We apologize for the confusion in the figure and will add clearer annotations in the revised version.
> >In in-context learning of LLM, how does the parameterization of this tool ensure its uniqueness? And how does it guarantee the generalizability of LLM tool invocations?
>
> In ParaTool's three-stage framework, each tool T_i is encoded as an independent set of LoRA parameters Θ_i. Regarding uniqueness, each tool's LoRA module is trained independently on its own dedicated dataset D_i during the Pre-training stage. The document-free training format removes tool documentation entirely, forcing the model to internalize the tool's specific functional semantics (parameter names, type constraints, invocation format, etc.) into its dedicated parameters rather than relying on in-context information. Since different tools' LoRA parameters are isolated from each other in the parameter space with no sharing, each set of parameters naturally encodes the unique knowledge of its corresponding tool.
>
> Regarding generalizability, ParaTool's soft tool selection mechanism enables flexible combination of these parametric tools: the gating network dynamically computes aggregation weights based on the user query and performs weighted fusion of multiple relevant tools' parameters, allowing the model to handle diverse query scenarios and multi-tool collaboration. Stage 3 joint fine-tuning further enhances generalizability by optimizing tool parameters under various aggregation weight configurations, teaching the model to correctly distinguish and invoke each tool across different parameter mixture settings.
>
> In summary, ParaTool ensures the uniqueness of each tool's representation through independent parameterization, while achieving generalizability of tool invocations across different scenarios through soft combination and joint fine-tuning.

---

> > ### Author Rebuttal · Reviewer_ZGJR · 2026-04-04
> >
> > So far the author has been able to solve my problem very well, but I still keep my score.

---

### Official Review · Reviewer_svyV · 2026-03-11

**Soundness:** 3
**Presentation:** 3
**Significance:** 2
**Originality:** 3
**Overall Recommendation:** 3
**Confidence:** 4

**Summary:**

This paper introduces ParaTool, a parameter-based framework designed to facilitate tool calling from an extensive library of candidates. The proposed method operates in three distinct stages: Parametric tool pre-training, soft tool selection, and parametric tool fine-tuning. Evaluations conducted on the Stable ToolBench and BFCL benchmarks indicate that ParaTool outperforms traditional In-Context Learning (ICL) baselines while simultaneously lowering computational overhead.

**Compliance With Llm Reviewing Policy:**

Affirmed.

**Key Questions For Authors:**

See Weakness

**Limitations:**

See Weakness

**Strengths And Weaknesses:**

### Strengths

* The paper addresses a critical and timely problem in the field—scaling tool-use capabilities to handle massive toolsets.
*  The authors provide both an empirical evaluation of the method and a supporting theoretical analysis.

### Weaknesses

*  The paper feels like an assembly of existing techniques rather than a cohesive new philosophy. It lacks deep insight into *why* this specific combination is necessary for the target problem. This is reflected in the standard ablation studies, which offer little in the way of novel observations or "lessons learned."
*  The evaluation relies on comparisons with classic or basic baselines. To truly demonstrate the value of ParaTool, the authors should compare it against more modern, state-of-the-art parameter-based tool-calling methods. Without this, the performance gains are difficult to contextualize.
* While the paper includes proofs, they are largely standard applications of existing theory. There are no significant technical highlights or novel theoretical frameworks introduced, making the "theoretical contribution" claim feel overstated.

---

> ### Author Rebuttal · Authors · 2026-03-31
>
> Thank you for your valuable comments and thorough feedback.
> >The paper feels like an assembly of existing techniques rather than a cohesive new philosophy. It lacks deep insight into why this specific combination is necessary for the target problem. This is reflected in the standard ablation studies, which offer little in the way of novel observations or "lessons learned."
>
> We want to clarify that ParaTool is not merely an assembly of existing techniques. It introduces a new paradigm that moves tool calling from the context space to the parameter space, enabling LLMs to use tools through dynamically integrated parameterized modules rather than in-context documents or examples. The three stages of ParaTool are inevitable design decisions, each addressing a distinct technical challenge of this new paradigm: tool-specific LoRA modules are necessary to avoid knowledge interference between tools, as demonstrated by the poor performance of the Global Parameterization baseline; the soft tool selection mechanism is essential to mitigate the impact of gating errors, since hard selection would irreversibly propagate such errors to downstream generation, evidenced by the 17% performance drop on Live Parallel Multiple compared with ParaTool; and the Parametric Tool Fine-tuning stage is crucial for aligning the independently optimized parameters from pre-training with the inference-time soft combination pattern, as evidenced by the 27.4% performance drop when this stage is removed. These results collectively demonstrate that each component plays an indispensable role.
> >The evaluation relies on comparisons with classic or basic baselines. To truly demonstrate the value of ParaTool, the authors should compare it against more modern, state-of-the-art parameter-based tool-calling methods. Without this, the performance gains are difficult to contextualize.
>
> We want to clarify that our baselines cover three major categories of methods: (1) ICL methods (Context+Docs, Context+Docs & Examples), representing the most widely used tool-calling paradigm; (2) Fine-tuning-based methods (Global Parameterization), which fine-tune LLMs on tool-calling trajectories. Existing fine-tuning-based approaches primarily differ in their data generation strategies, while sharing similar underlying training procedures [1]. To ensure a fair comparison, we train the "Global Parameterization" baseline on exactly the same trajectory dataset as our ParaTool. This baseline employs a single LoRA module with a larger rank, trained jointly on all tool-calling data, thereby encoding the information of all tools simultaneously. It serves as a strong representative of the fine-tuning-based paradigm; (3) Tool selection methods (EmbSim, ToolRetriever, Re-Invoke), representing advanced tool selection strategies. ParaTool is to move tool usage from the context space to the parameter space. Under this new paradigm, ParaTool achieves significant performance gains over all these baselines, demonstrating the effectiveness of our approach.
>
> [1] Qu C et al., Tool learning with large language models: A survey, 2025.
> >While the paper includes proofs, they are largely standard applications of existing theory. There are no significant technical highlights or novel theoretical frameworks introduced, making the "theoretical contribution" claim feel overstated.
>
> We would like to clarify the contributions of our theoretical analysis. Our theoretical analysis aims to answer a critical design question: "Is Soft Tool Selection necessary?" Specifically, in Theorem 3.6 (Gradient Norm Bound), we investigate the relationship between the aggregation weight distribution and the gradient norm, revealing how the ℓ₂ norm of the weights modulates the model's robustness through the gradient alignment coefficient α_g. Intuitively, one might assume that concentrated weights (hard selection) would yield more precise tool representations, yet the theory shows that soft combination achieves a larger robustness radius by reducing the gradient norm. Furthermore, Corollary 3.7 (Robustness Advantage of Soft Combination) directly guided our design decision to adopt soft selection over hard selection. As confirmed by our ablation studies, hard selection leads to a consistent performance drop across all categories, with the most pronounced degradation of nearly 17% observed in the Live Parallel Multiple category.

---

> > ### Author Rebuttal · Reviewer_svyV · 2026-04-03
> >
> > I disagree with the authors that "It introduces a new paradigm that moves tool calling from the context space to the parameter space". This seems overclaim the contribution of this paper, as previous works explore the parameter finetuning to enhance the LLM's tool calling ability. What's more, Global Parameterization is a very naive baseline. Little can say by surpassing such naive baseline.

---

> > > ### Author Response · Authors · 2026-04-06
> > >
> > > Thank you for this important comment. Our intention was not to claim that ParaTool is the first method to use parameter fine-tuning for tool calling, nor that it is simply a newer fine-tuning approach for enhancing general tool-use ability. What we mean by a paradigm shift is more specific: previous tuning-based methods improve general tool-calling ability, but they often still **rely on in-context tool documentation** to provide the model with the specific details of previously seen tools. However, just as illustrated in paper's Figure 1, as we introduce more and more tool information into the context, the model’s tool-calling capability paradoxically declines. The extensive context not only incurs high inference latency and memory overhead but also exacerbates the risk of hallucination, making it difficult for the model to precisely capture effective content within verbose prompts. In this sense, the shift we emphasize is not from “no parameter tuning” to “parameter tuning”, but from representing tool-specific knowledge primarily in the context space to representing it through tool-specific parameter modules that can be dynamically composed at inference time. Therefore, ParaTool should not be viewed as another fine-tuning method for generally improving tool-calling ability, but rather as a framework for reducing reliance on in-context tool documentation by parameterizing tool-specific knowledge itself.
> > >
> > > To further alleviate your concern, we add new comparisons with ToolLLaMA[1], ToolAlign[2], and iTool[3] on the Qwen2.5-14B model. To encourage the model to internalize tool knowledge, we used the same document-free setting as ParaTool during data synthesis. As shown in the table above, ParaTool consistently outperforms these methods across the five categories of BFCL. The advantage is especially pronounced on the more challenging categories, such as Live Parallel and Live Parallel Multiple. This demonstrates our opinion again that while tuning-based methods improve general tool-calling capabilities, they often fail to effectively internalize the specific details of previously seen tools, thereby retaining a dependency on in-context documentation. In the revised version, we will add these results to support our empirical claims.
> > >
> > > | Method                   | Parallel | Multiple | Parallel Multiple | Live Parallel | Live Parallel Multiple |
> > > |:-------------------------|:--------:|:--------:|:-----------------:|:-------------:|:----------------------:|
> > > | ToolLLaMA                |74.50     |93.50     |85.50              |60.42          |37.50                   |
> > > | ToolAlign                |77.83     |86.00     |59.50              |70.83          |34.72                   |
> > > | iTool                    |75.50     |86.00     |63.00              |70.83          |45.83                   |
> > > | Global Parameterization  |74.50     |86.00     |59.33              |60.42          |31.94                   |
> > > | **ParaTool**             |**95.00** |**95.50** |**92.00**          |**93.75**      |**87.50**               |
> > >
> > > [1] Qin Y. et al, Toolllm: Facilitating Large Language Models to Master 16000+ Real-World APIs, ICLR 2024
> > >
> > > [2] Chen Z.-Y. et al, Towards Tool Use Alignment of Large Language Models, ACL 2024
> > >
> > > [3] Zeng Y. et al, iTool: Reinforced Fine-tuning with Dynamic Deficiency Calibration for Advanced Tool Use, EMNLP 2025

---

### Official Review · Reviewer_Gxoi · 2026-03-12

**Soundness:** 2
**Presentation:** 3
**Significance:** 2
**Originality:** 2
**Overall Recommendation:** 3
**Confidence:** 4

**Summary:**

The paper introduces ParaTool, a parameter-based integration framework that converts tool representation into a loadable set of parameters. The three-stage pipeline includes pre-training each tool into a LoRA module, training a gating network to predict aggregation weights based on the query and history, and fine-tuning tool parameters end-to-end. Experiments on Stable ToolBench and BFCL show that ParaTool achieves strong performance gains and reduces computational cost compared to ICL-based methods.

**Compliance With Llm Reviewing Policy:**

Affirmed.

**Final Justification:**

Thanks for the additional experiments on Qwen2.5-14B and comparisons with ToolLLaMA, ToolAlign, and iTool. However, key concerns remain.
1. The performance margins over tuning-based baselines are implausibly large (e.g., ParaTool outperforms ToolLLaMA by +33.33% on Live Parallel and +50% on Live Parallel Multiple, and outperforms ToolAlign by +22.92% and +52.78% respectively), which is hard to justify given that LoRA-based methods are generally capacity-constrained relative to full fine-tuning.
2. The reported BFCL results (e.g., Multiple 95.50, Live Parallel Multiple 87.50) substantially exceed publicly available leaderboard results from the top-ranked model (Claude-Opus-4-5-20251101 (FC): Multiple 78.16, Live Parallel Multiple 75), raising serious doubts about evaluation validity.
3. The rebuttal does not clarify whether all baselines share the same LLM backbone and data splits.
These unresolved concerns significantly undermine the credibility of the empirical claims, and I would like to maintain my score.

**Key Questions For Authors:**

1. In Figure 3, ParaTool without fine-tuning achieves low accuracy. In Figure 11, the gating accuracy is relatively low (e.g., 65% on BFCL), but the final action accuracy is much higher. This suggests that the LLM can correct gating errors. Could you provide more analysis on why this correction occurs? Is this primarily due to residual information, or the reasoning capabilities of the model?
2. Recent works, such as ToolGen[3], also transforms tool representation from in-context to model parameters. I was wondering if the authors could discuss how ParaTool compares to such strategies. Specifically, how does ParaTool's dynamic LoRA aggregation differ from or complement ToolGen's virtual token approach? Adding a brief comparative discussion in the Related Work would help clarify the paper's unique contribution.

**Limitations:**

Yes

**Strengths And Weaknesses:**

Strengths
1. The paper proposes an efficient tool representation method by internalizing tool knowledge into model parameters, handling the need for long tool documentations in the context.
2. ParaTool consistently achieves state-of-the-art performance across all baselines on both the Stable ToolBench and BFCL benchmarks, showing its tool-use capability and computation efficiency.
3. The FLOPs analysis shows significant reduction in computational cost(up to 92.22% on Stable Toolbench and 94.45% on BFCL) compared to ICL-based methods.

Weaknesses
1. The paper compares ParaTool against ICL baselines and retrieval-based tool selectors, but it does not include comparisons with other parameter-efficient tool learning methods. For example, methods like ToolLLama[1] or ToolAlign[2] fine-tune LLMs on tool-calling data, exhibiting remarkable tool-calling capabilities.

2. ParaTool requires pre-training a LoRA module for each tool, which limits it to a fixed set of tools. It is not scalable to large and dynamically changing tool sets, and it cannot generalize to unseen tools without retraining.

3. The experiments are conducted on Llama-3.1-8B and Qwen2.5-7B, which are relatively small. It is unclear whether the gains hold for larger, instruction-tuned models (e.g., Qwen2.5-14B) that already exhibit strong tool-use capabilities.

---

> ### Author Rebuttal · Authors · 2026-03-31
>
> Thank you for your valuable comments and thorough feedback.
> >The paper doesn't include comparisons with parameter-efficient tool learning methods, such as ToolLLama or ToolAlign.
>
> ToolLLama and ToolAlign belong to tuning-based methods that fine-tune LLMs to improve general tool-calling capabilities. For a fair comparison, we need to evaluate ParaTool against these methods on the same trajectory dataset. To this end, we have included the baseline "Global Parameterization" as a strong representative of tuning-based methods, which utilizes a single LoRA module (with larger rank) trained on the entire dataset to encode the knowledge of all tools. The average performance of Global Parameterization is 33% lower than ParaTool on the BFCL benchmark. The core issue is that compressing all tool knowledge into a shared parameter space makes it difficult for the model to disentangle individual tool functionalities, leading to catastrophic forgetting and hallucinations. We will discuss the relationship between ParaTool and ToolLLaMA/ToolAlign in the revised version.
> >ParaTool cannot scale to large and dynamically changing tool sets, and it cannot generalize to unseen tools without retraining.
>
> Each tool is represented by an independent LoRA module, which makes it possible to add a new tool by training only its corresponding module rather than retraining the entire system. To validate this, we conduct an experiment on Qwen2.5-14B: we first parameterize all tools from BFCL's Multiple category, then select a 5% subset from Live Multiple, parameterize the corresponding new tools, and incorporate them into the existing toolset. The gating network is retrained to accommodate the expanded toolset, and since it is just an MLP, this retraining remains low-cost. On this subset, ParaTool achieves 85% accuracy versus 80% for Context+Docs\&Examples. Moreover, performance on the original Multiple category is unaffected (95.5→95.5).
>
> For unseen tools, it is indeed not feasible in the current version. As noted in our conclusion, we plan to address unseen tools via meta-learning(e.g., hypernetworks) in future work.
> >It isn't clear whether ParaTool can get gains on larger language models.
>
> | Method                   | Parallel | Multiple | Parallel Multiple | Live Parallel | Live Parallel Multiple |
> |:-------------------------|:--------:|:--------:|:-----------------:|:-------------:|:----------------------:|
> | Context+Docs             |93.50     |93.50     |87.50              |56.25          |75.00                   |
> | Context+Docs&Examples    |94.00     |95.00     |90.00              |81.25          |79.17                   |
> | Global Parameterization  |86.00     |74.50     |59.33              |60.42          |31.94                   |
> | Parameter+Embsim         |          |91.50     |64.50              |               |62.50                   |
> | Parameter+Tool Retriever |          |92.50     |67.00              |               |58.33                   |
> | Parameter+Reinvoke       |          |92.50     |68.50              |               |70.83                   |
> | **ParaTool**             |**95.00** |**95.50** |**92.00**          |**93.75**      |**87.50**               |
>
> We further evaluate ParaTool on the Qwen2.5-14B model. As shown in the table, ParaTool outperforms all baseline methods across every category, with an average relative improvement of 5.94% over the strongest baseline (Context+Docs&Examples). The advantage is pronounced on the more challenging categories: ParaTool surpasses the best baseline by 15.38% on Live Parallel and 10.53% on Live Parallel Multiple. Due to the limited rebuttal period, Live Multiple results are in progress and will be included in the final version.
> >Why does the final action accuracy exceed the gating accuracy? Is this due to residual information or the model's reasoning capabilities?
>
> Firstly, the soft selection mechanism allows the correct tool's parameters to still influence the model's behavior even when not selected as the top choice. Secondly, the model can not inherently leverage such residual information, as shown by low accuracy without Stage 3. To teach the model to exploit this residual signal, Stage 3 fine-tuning aligns the training process with the inference process, enabling the model to learn how to utilize the blended parameter space produced by soft gating.
> >How does ParaTool differ from ToolGen's virtual token approach?
>
> ToolGen is designed to solve the tool selection problem by learning a token-level representation for each tool. But tool calling involves not only selecting the correct tool but also correctly invoking it with the right parameters and format. ToolGen still relies on in-context learning to teach the model how to use the selected tool. In contrast, ParaTool transform the entire tool calling process into the parameter space, which allow the model to invoke tools directly through the learned parameters. We will add this comparison in the revised Related Work.

---

> > ### Author Rebuttal · Reviewer_Gxoi · 2026-04-03
> >
> > Thanks for your responses. The rebuttal addresses most of my concerns, particularly regarding scalability experiments and larger model evaluations. However, "Global Parameterization" (a single LoRA for all tools) is more like an ablation of your design, not a strong representative of established tuning-based methods like ToolLLaMA or ToolAlign. The unseen tool generalization limitation is a real and significant constraint of the current framework.

---

> > > ### Author Response · Authors · 2026-04-06
> > >
> > > We sincerely thank you for your thoughtful follow-up and for highlighting these important concerns. Your comments have helped us further improve the paper. To further alleviate your concern, we add new comparisons with ToolLLaMA[1], ToolAlign[2], and iTool[3] on the Qwen2.5-14B model. To encourage the model to internalize tool knowledge, we used the same document-free setting as ParaTool during data synthesis. As shown in the table above, ParaTool consistently outperforms these methods across the five categories of BFCL. The advantage is especially pronounced on the more challenging categories, such as Live Parallel and Live Parallel Multiple. This demonstrates our opinion again that while tuning-based methods improve general tool-calling capabilities, they often fail to effectively internalize the specific details of previously seen tools, thereby retaining a dependency on in-context documentation. In the revised version, we will add these results to support our empirical claims.
> > >
> > > | Method                   | Parallel | Multiple | Parallel Multiple | Live Parallel | Live Parallel Multiple |
> > > |:-------------------------|:--------:|:--------:|:-----------------:|:-------------:|:----------------------:|
> > > | ToolLLaMA                |74.50     |93.50     |85.50              |60.42          |37.50                   |
> > > | ToolAlign                |77.83     |86.00     |59.50              |70.83          |34.72                   |
> > > | iTool                    |75.50     |86.00     |63.00              |70.83          |45.83                   |
> > > | Global Parameterization  |74.50     |86.00     |59.33              |60.42          |31.94                   |
> > > | **ParaTool**             |**95.00** |**95.50** |**92.00**          |**93.75**      |**87.50**               |
> > >
> > > As for the limitation on unseen-tool generalization, we agree that the current version of ParaTool does not support zero-shot generalization to truly unseen tools. What ParaTool does support is incremental extension to newly added tools: a new tool can be incorporated by training its corresponding LoRA module and updating the gating network, which is a lightweight MLP and therefore relatively inexpensive to retrain. Our scalability experiment in the last round of rebuttal suggests that this practical extensibility to newly introduced tools may alleviate the challenge of generalizing to truly unseen tools to some extent. At the same time, we acknowledge that this is not an optimal solution to unseen-tool generalization, and we will state this limitation more explicitly in the final version.
> > >
> > > [1] Qin Y. et al, Toolllm: Facilitating Large Language Models to Master 16000+ Real-World APIs, ICLR 2024
> > >
> > > [2] Chen Z.-Y. et al, Towards Tool Use Alignment of Large Language Models, ACL 2024
> > >
> > > [3] Zeng Y. et al, iTool: Reinforced Fine-tuning with Dynamic Deficiency Calibration for Advanced Tool Use, EMNLP 2025

---

### Official Review · Reviewer_enSE · 2026-03-12

**Soundness:** 3
**Presentation:** 4
**Significance:** 3
**Originality:** 3
**Overall Recommendation:** 5
**Confidence:** 4

**Summary:**

ParaTool proposes a different paradigm for tool-augmented LLMs, replacing in-context tool documentation with dedicated parametric representations for each tool. The motivation is  twofold: ICL-based approaches suffer from both long context overhead and could be prone to hallucinations as examples accumulate while tuning-based methods improve general instruction following but fail to internalize specific tool details. ParaTool addresses this via three stages: (1) parametric tool pre-training,  which encodes each tool's knowledge into an independent LoRA-style adapter; (2) soft tool selection using a gating network to dynamically aggregate relevant tool parameters; and (3) parametric tool fine-tuning that jointly optimizes tool parameters to align training and inference. Experiments on Stable ToolBench and BFCL show +9.71% and +4.22% accuracy improvements respectively, with FLOPs reductions of 92-94% compared to ICL baselines.

**Compliance With Llm Reviewing Policy:**

Affirmed.

**Final Justification:**

The authors addressed my concerns in the rebuttal. I think there are some missing baselines that other reviewers discussed, but on my side things are ok. I am sticking to my score.

**Key Questions For Authors:**

- It would be interesting to add an experiment to measure  if ParaTool  causes forgetting of the base LLM's general capabilities, does it degrade the base LLM's capabilities less than standard fine-tuning?
- How  would the method scale to very large tool suites? I understand that it would probably be cheaper than passing a long explanation of all tools in the suite in context but how can you ensure that there is enough data so all parameters in ParaTool are adequately trained?

**Limitations:**

- No, the authors do not have a dedicated "Limitations" section, and they do not substantively discuss the limitations of their own approach. Some noteworthy things that are worth discussing would be:
  - Failure modes or cases where their parametric tool encoding underperforms
  - Scalability
  - Cost/overhead of the three-stage fine-tuning pipeline
  - Generalization to tools not seen during training (only mentioned as future work, not framed as a current limitation)

**Strengths And Weaknesses:**

Strengths:

Originality and significance: the authors propose an interesting solution, namely, adding tool-specific parameters plus soft selection, in order to avoid the giant context when passing tools documentation and examples during ICL. The authors empirically show that accuracy peaks then drops with more in-context examples.

Soundness: very thorough ablations to justify their design choices:  i) pre-training with tool-params only, ii) top-1 instead of soft selection, etc. They compare against reasonable baselines such as ICL, LORA fine-tuning, among others and they further train two OSS models in order to showcase their methodology. The authors also present a theoretical justification why  transforming the heavy processing of
verbose documents into lightweight low-rank computations is computationally cheaper.

Presentation: the paper is well written and easy to read, the motivation is stated clearly throughout.

Weaknesses:

There is no discussion about potential failure modes.

---

> ### Author Rebuttal · Authors · 2026-03-31
>
> Thank you for your valuable comments and thorough feedback.
> > It would be interesting to add an experiment to measure if ParaTool causes forgetting of the base LLM's general capabilities, does it degrade the base LLM's capabilities less than standard fine-tuning?
>
> ParaTool is designed with a natural advantage in preventing capability degradation. Specifically, the tool-specific LoRA modules in ParaTool are dynamically loaded and activated by the gating network only when needed for tool invocation. When all tools are completely unloaded, the base model reverts to its original parameters. In contrast to global fine-tuning methods that permanently alter all model parameters and inevitably degrade performance on other tasks, ParaTool fundamentally preserves the base model's general capabilities by design.
> > How does ParaTool scale to very large tool suites, and how can you ensure sufficient training data for all parameters?
>
> In ParaTool, each tool corresponds to an independent LoRA module. This characteristic allows for incremental expansion of the toolset. To validate this potential, we conduct a simple experiment based on Qwen2.5-14B. We first follow the standard ParaTool procedure to parameterize all tools from BFCL's Multiple category. We then select 5% subset from Live Multiple category, parameterize the corresponding new tools, and incorporate them into the existing parameterized toolset. The gating network, originally trained on the Multiple category, is subsequently retrained to accommodate the expanded toolset. Notably, since the gating network is just an MLP, this retraining process remains relatively low-cost. On this subset of Live Multiple, ParaTool achieves 85% accuracy, compared to 80% for Context+Docs\&Examples. Moreover, we re-evaluate on the original Multiple category and find that incorporating new tools into the parameterized toolset does not degrade performance (95.5->95.5).
>
> To ensure sufficient training data for each tool, we design a systematic data generation pipeline (Appendix C): We first use an LLM to generate a sufficient number of atomic question-answer examples for each tool. These atomic examples are then programmatically combined via scripts into coherent multi-turn tool-calling trajectories. To further increase trajectory complexity, we augment each example with a candidate tool set that includes not only the target tool but also distractor tools retrieved based on Jaccard similarity of their schemas.
> > The authors do not have a dedicated "Limitations" section, and they do not discuss the limitations of their own approach (failure modes, cost/overhead of the three-stage fine-tuning pipeline, and generalization to unseen tools).
>
> We thank the reviewer for this suggestion. We acknowledge that the original manuscript lacked a Limitations section and will add one in the revised version. We briefly discuss these limitations below:
>
> 1. Failure modes. For efficiency, ParaTool uses a lightweight MLP-based gating network for soft tool selection. When the query is highly complex or candidate tools have very similar schemas, the gating network may assign insufficient weight to the correct tool, limiting the model's access to the right parametric knowledge. Future work could adopt an LLM-based tool selection mechanism for more accurate tool weighting.
>
> 2. Cost/overhead of the three-stage pipeline. Stage 1 and Stage 3 are both LoRA-based (rank $r=16$, $\alpha=64$, targeting FFN weights only), trained for 1–3 epochs. Stage 2 trains only a 3-layer MLP (hidden dim 512) for 3 epochs without updating any LLM parameters (Appendix D). The entire pipeline is built on parameter-efficient fine-tuning with modest computational cost.
>
> 3. Generalization to unseen tools. ParaTool requires each tool to be parametrized during training, so it cannot leverage tools not yet encoded into LoRA modules. As discussed in our conclusion, we plan to explore meta-learning approaches (e.g., hypernetworks) to translate tool documentation directly into parametric representations, enabling rapid adaptation to unseen tools without full retraining.

---

> > ### Author Rebuttal · Reviewer_enSE · 2026-04-03
> >
> > My concerns have been addressed in the rebuttal.

---

### Decision · Program_Chairs · 2026-04-30

**Decision:**

Accept (regular)

**Comment:**

The paper introduces ParaTool, a framework that replaces in-context tool documentation with per-tool LoRA modules, dynamically combined at inference time via a gating network. The method has three stages: independent LoRA pre-training per tool, a gating network that predicts tool-aggregation weights from the query, and joint LoRA fine-tuning under the soft-weighted combination to align training with inference. At inference, the gating MLP produces softmax weights over candidate tools, the corresponding LoRAs are combined by weighted sum, and the LLM generates the tool call without tool documentation in context. Reported gains over ICL baselines are +9.71% on Stable ToolBench and +4.22% on BFCL with 92–95% FLOPs reduction, and a rebuttal-added Qwen2.5-14B experiment shows +5.94% average improvement over the strongest baseline.

The reviewers split (scores 5, 5, 3, 3) but concerns cluster around the strength of the tuning-based baseline. Reviewer_enSE and Reviewer_ZGJR were fully satisfied with the rebuttal, which added a Limitations section, an incremental tool-expansion experiment showing no degradation on previously parameterized tools, and a ToolGen comparison clarifying how ParaTool differs from prior tool-selection methods. Reviewer_Gxoi maintained a 3 on the specific concern that the paper does not compare against established tuning-based methods such as ToolLLaMA or ToolAlign. Reviewer_svyV maintained a 3 on a related but broader concern: that the "new paradigm" framing overclaims given prior parameter fine-tuning work for tool calling, and that Global Parameterization is a naive baseline whose defeat does not strongly support the paper's claims.

The AC finds the architectural contribution substantive and the practical picture strong. Per-tool parameterization with soft gating is a cleanly isolated design choice, and the ablations support each component, with removing Stage 3 causing a 27% drop and replacing soft selection with hard top-1 causing up to a 17% drop. Training cost per tool is modest, and the incremental-expansion experiment demonstrates new tools can be added without degrading existing ones. On Reviewer_Gxoi's concern, the authors' argument is defensible: Global Parameterization is intended to isolate the architectural comparison, and comparing against ToolLLaMA or ToolAlign at their published scales would answer a different question. An addition of a closer comparison to at least one established tuning-based method would nonetheless strengthen the empirical case.

The AC recommends acceptance. The paper's architectural contribution is clear, its ablations are thorough, and the practical properties support the claim that per-tool parameterization is a useful alternative to ICL-based tool calling. The authors should add the Limitations section committed in the rebuttal, discuss the relationship to ToolLLaMA and ToolAlign more explicitly in Related Work, include at least one direct comparison to an established tuning-based method in the final version.